# A²SG: Adaptive and Asymmetric Surrogate Gradients for Training Deep Spiking Neural Networks

## Abstract

Training deep spiking neural networks (SNNs) remains challenging due to sharp loss landscapes and temporal inconsistency caused by surrogate gradients. To address these challenges, we propose a unified framework: adaptive and asymmetric surrogate gradients ($A^2SG$). The adaptive gradients adjust an effective window for spatio-temporal adaptation, reducing spatial gradient variation and maintaining directional consistency of gradients over time. The asymmetric gradients reflect neuronal dynamics by assigning larger gradients to neurons with higher membrane potentials, and we prove that they yield lower variation than symmetric surrogates. Our analysis further establishes a direct connection between local gradient variation and the curvature of the loss landscape, providing a principled explanation for how $A^2SG$ promotes convergence to flatter minima and improves generalization. We conduct extensive experiments on diverse models, including CNN-based and Transformer-based SNNs, across various tasks such as image classification using both static and neuromorphic datasets, as well as segmentation. The results demonstrate that $A^2SG$ consistently improves accuracy and energy efficiency, establishing it as a general and reliable solution for training deep SNNs.

## 1 Introduction

Spiking neural networks (SNNs) have emerged as energy-efficient next-generation neural networks that operate based on spikes (Maass, 1997). In particular, by leveraging the low-power characteristics of SNNs and the superior learning capabilities of deep neural networks (DNNs), deep SNNs have shown the potential for energy-efficient artificial intelligence in various fields (Park et al., 2020; Kim et al., 2020b; 2022b; Yao et al., 2025). Recently, deep SNNs have been successfully applied to various applications, including image segmentation (Kim et al., 2022b; Lei et al., 2025), object detection (Kim et al., 2020b; Su et al., 2023), and language modeling (Bal & Sengupta, 2024; Xing et al., 2024), as well as to diverse model architectures, such as Transformers (Zhou et al., 2023; Yao et al., 2025). These rapid advancements have been largely driven by the adoption of gradient-based training with surrogate gradients (Wu et al., 2018; Neftci et al., 2019).

Despite their essential role in training deep SNNs, research on effective surrogate gradient functions remains limited, contributing to the performance gap between DNNs and deep SNNs. Several studies have attempted to mitigate the mismatch between surrogate and true gradients by adaptively adjusting surrogate functions. However, existing approaches have predominantly focused on gradient sparsity for adaptation (Lian et al., 2023; Lin et al., 2023) or impose substantial computational overhead (Li et al., 2021), restricting the training performance or hindering practical deployment. Furthermore, few studies have designed surrogate functions that take into account the impact of surrogate gradients on generalization performance.

In this work, we introduce adaptive and asymmetric surrogate gradients ($A^2SG$) to enhance the training of deep SNNs. The adaptive component leverages spatio-temporal adaptation, dynamically adjusting the surrogate gradient window to suppress spatial fluctuations of gradients and align their directions across timesteps. The asymmetric component allocates larger gradients to neurons with greater membrane potential, effectively prioritizing those closer to firing and promoting convergence to flatter minima. These designs are motivated by our theoretical analysis, which shows that a

larger variation in local gradient leads to sharper loss landscapes. Moreover, we demonstrate that the proposed asymmetric surrogate gradient exhibits lower gradient variation compared to its symmetric counterparts. In addition, we highlight temporal model collapse, which is caused by misaligned gradients across timesteps, as one of the obstacles for stable learning, motivating the need for spatio-temporal adaptation. By combining these, we establish a unified strategy that stabilizes optimization, promotes convergence to flatter minima, and improves generalization. We validate our proposed approaches through extensive experiments on both static and neuromorphic datasets, spanning convolutional neural networks (CNNs), Transformer-based models, and segmentation tasks. Across all benchmarks, $A^2SG$ achieves consistent gains in accuracy and energy efficiency, highlighting its effectiveness as a general solution for reliable deep SNN training.

## 2 RELATED WORK

### 2.1 TRAINING DEEP SPIKING NEURAL NETWORKS

Recent studies have introduced deep SNN achieving both high performance and energy efficiency (Tavanaei et al., 2019). In these architectures, leaky integrate-and-fire (LIF) neurons are widely used for their computational simplicity and biological plausibility (Eqs. A1- A3). Deep SNNs have been applied to various tasks such as image classification (Hu et al., 2021; Fang et al., 2021a), object detection (Kim et al., 2020a;b), semantic segmentation (Kim et al., 2022b; Lei et al., 2025), and Transformer-based models (Zhou et al., 2023; Yao et al., 2025). Recent studies have adopted direct training based on spatio-temporal backpropagation (STBP) with surrogate gradients (Wu et al., 2018), as given in Eqs. A4 and A5. This method enables efficient training with fewer time steps, yet a performance gap remains compared to conventional DNNs. To mitigate this gap, several studies have focused on addressing the gradient mismatch problem arising from the adoption of surrogate gradients (Li et al., 2021; Lian et al., 2023). However, studies on gradient consistency during training have remained relatively limited. Especially, in STBP, parameter updates are obtained by aggregating gradient contributions from all timesteps, and inconsistent temporal gradients can generate conflicting signals, a phenomenon we term *temporal gradient confusion*. This inconsistency hinders stable optimization and degrades learning performance, highlighting the need for strategies that explicitly mitigate it.

### 2.2 SURROGATE GRADIENTS IN SPATIO-TEMPORAL BACKPROPAGATION (STBP)

Surrogate gradients have been employed to address the non-differentiability of spiking function (Eq. A2) during error backpropagation. Although the adoption of surrogate gradients has dramatically improved the performance of deep SNNs, their use remains limited by inconsistencies with the true gradients. To improve the learning performance, several studies have focused on adjusting the distribution of the membrane potential during training (Guo et al., 2022; 2023a;b; Zhao et al., 2025). Various regularization strategies have been employed, such as maximizing the information within the membrane potential (Guo et al., 2022) or minimizing quantization errors induced by the spike function (Guo et al., 2023a). In addition, batch normalization (Guo et al., 2023b) and KL loss (Zhao et al., 2025) were applied to mitigate the inter-batch and temporal discrepancies in the membrane potential distribution, respectively. While these studies improved performance by adjusting the membrane potential distribution, they did not fundamentally address the performance degradation inherent to surrogate gradients, highlighting the essential need for advancements in surrogate gradient design.

### 2.3 IMPROVING SURROGATE GRADIENTS DESIGN FOR DEEP SNNS

Most surrogate gradients adopt static and symmetric function shapes to approximate the Dirac delta, which represents the derivative of the spiking function ($\frac{\partial s[t]}{\partial u[t]}$). These functions preserve a constant area within the effective window $[V_{\text{th}} - \beta, V_{\text{th}} + \beta]$, with representative examples being the boxcar (*BOX*) and triangle (*TRI*) functions (Eqs. A6-A7.) To mitigate vanishing gradients, Guo et al. (2024) proposed directly delivering gradients to shallow layers, though gradient mismatch remains unresolved. Beyond static functions, adaptive strategies have been explored to improve training further (Li et al., 2021; Lian et al., 2023; Lin et al., 2023; jia). Dspike (Li et al., 2021) employs finite-difference gradients to align surrogates with true gradients via cosine similarity, though its

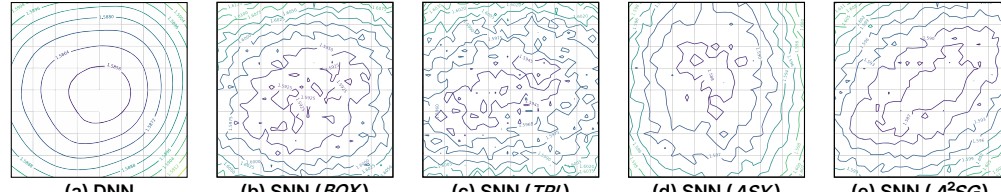

(a) DNN     (b) SNN ($BOX$)     (c) SNN ($TRI$)     (d) SNN ($ASY$)     (e) SNN ($A^2SG$)

Figure 1: Visualization of the loss landscape in the Conv1 layer of VGG16 trained on CIFAR10. The results demonstrate that our adaptive surrogate gradient leads to a flatter loss landscape compared to conventional surrogate gradients.

high computational cost limits scalability. To reduce this overhead, other studies propose adjusting the effective window width to regulate gradient sparsity during training (Lian et al., 2023; Lin et al., 2023; jia). However, most of these approaches have focused on sparsity control to avoid gradient vanishing or explosion, relying on sparsity as an indirect indicator of learning quality. Moreover, existing efforts have primarily concentrated on symmetric functions that mimic differential operators, while the impact of function shape itself remains underexplored.

## 2.4 FLAT MINIMA AND GENERALIZATION

Although gradient-based direct learning has significantly improved the performance, generalization remains a critical challenge for deep SNNs. Numerous studies have reported that models converging to flat minima tend to achieve better generalization (Hochreiter & Schmidhuber, 1997; Keskar et al., 2017; Chaudhari et al., 2019). Flatness is commonly assessed via the Hessian spectrum (Ghorbani et al., 2019), but direct Hessian computations are intractable for large models. As a practical alternative, the Fisher information matrix (FIM) (Eq. A8) is frequently used, since its eigenvalues are known to capture the curvature of the loss landscape, with smaller values indicating flatter minima (Liao et al., 2018; Karakida et al., 2019; Martens, 2020; Kim et al., 2022a). In addition to measurement, several training procedures aim to encourage convergence to flat minima. These include entropy-based biasing toward wide valleys (Chaudhari et al., 2019) and curvature-aware updates using FIM-based criteria (Kim et al., 2022a). Related studies have also linked sharp minima to large-batch training and poor generalization (Keskar et al., 2017), further motivating flatness-oriented training strategies. Overall, prior works suggest that guiding optimization toward flat minima is an effective approach for improving generalization.

## 3 SHARP LOSS LANDSCAPE FROM SURROGATE GRADIENT LEARNING

Deep SNNs trained with surrogate gradients tend to converge to sharper loss landscapes than DNNs. The flatness of the loss landscape is characterized by its curvature, quantified by the Hessian $\mathbf{H} = \nabla_{\mathbf{w}}^2 L(\mathbf{w})$, with respect to the parameter vector $\mathbf{w}$. For clarity, we analyze the second derivative of the loss with respect to a single weight, which corresponds to a diagonal entry of the Hessian. By the chain rule, the second derivative of the loss with respect to a weight $w$ can be described as

$$\frac{\partial^2 L}{\partial w^2} = \frac{\partial^2 L}{\partial \phi^2}(\phi'(u)x)^2 + \frac{\partial L}{\partial \phi}\phi''(u)x^2, \tag{1}$$

where $x$ denotes the pre-synaptic activation and $u = wx$. For DNNs with an activation function $\phi(u)$, assume the first and second derivatives are bounded, i.e., $|\phi'(u)| \leq c_1$ and $|\phi''(u)| \leq c_2$ for finite constants $c_1$ and $c_2$, which yields

$$\left|\frac{\partial^2 L}{\partial w^2}\right|_{DNN} = \mathcal{O}(x^2). \tag{2}$$

For deep SNNs, a surrogate gradient function $f(u)$ introduces an effective window of width $\beta$ to approximate the non-differentiable spike function. As shown in Sec. A.5, any symmetric surrogate with fixed area satisfies

$$\|H'\|_\infty = \|f\|_\infty = \Omega(\beta^{-1}) \quad \text{and} \quad \|H''\|_\infty = \|f'\|_\infty = \Omega(\beta^{-2}). \tag{3}$$

Substituting them into the chain-rule expansion (Eq. 1) yields

$$\left|\frac{\partial^2 L}{\partial w^2}\right|_{SNN} = \Omega(\frac{x^2}{\beta^2}). \tag{4}$$

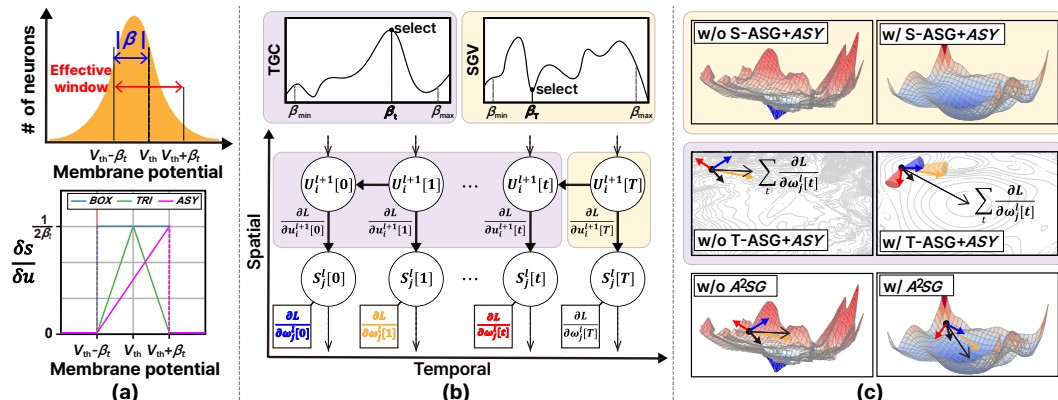

Figure 2: Overview of the proposed $A^2SG$ framework. (a) Effective window (red) is modulated by the parameter $\beta$ (blue); three shapes (*BOX*, *TRI*, and *ASY*) are shown. (b) Temporal adaptive surrogate gradient (T-ASG) selects $\beta_t$ that maximizes temporal gradient consistency (TGC). While Spatial adaptive surrogate gradient (S-ASG) selects $\beta_t$ that minimizes spatial gradient variation (SGV). (c) S-ASG+*ASY* promotes flat minima, T-ASG+*ASY* stabilizes gradient directions, and $A^2SG$ achieves robust convergence by combining both.

Most prior works adopt a *narrow effective window* ($\beta < 1$) to better approximate the Dirac delta around the threshold, which empirically improves training stability and convergence (Wu et al., 2018; Neftci et al., 2019). Under this condition, Eq. 4 shows that the Hessian magnitude is amplified by a factor of $1/\beta^2$ in surrogate-trained SNNs, indicating a sharper loss landscape than DNNs. In addition, the binary and temporally sparse nature of spikes concentrates gradients and increases their variation, further sharpening the landscape, as shown in Fig. 1. Consistent with this analysis, *TRI* yields a sharper loss landscape than *BOX* because, under area normalization, its steeper slopes imply larger curvature. This is derived in Sec. A.6 and confirmed empirically through Fig. 1-(b) and (c).

# 4   $A^2SG$: ADAPTIVE AND ASYMMETRIC SURROGATE GRADIENTS

We introduce $A^2SG$ to address two limitations in surrogate-based SNN training: sharp loss landscapes and temporal gradient confusion. The adaptive component consists of two policies: spatial and temporal adaptations to address sharp loss landscapes and temporal gradient confusion, respectively. The spatial adaptation adjusts the surrogate effective window to reduce the variance of the local gradient ($\frac{\partial L}{\partial u}$), encouraging convergence to flatter minima. The temporal adaptation aligns per-timestep local gradients to mitigate temporal inconsistency. We demonstrate that the dispersion of local gradients affects the curvature of the loss landscape and that temporal gradients are formed by aggregating these local gradients. Thus, adjusting $\beta$ during training effectively reduces spatial variability and enhances temporal alignment, improving generalization under non-stationary dynamics.

In addition, we analyze how the functional shape of the surrogate affects local gradient variation and introduce an asymmetric surrogate that considers neuronal dynamics. By allocating larger gradients to neurons with greater accumulated membrane potential, the asymmetric form further suppresses the gradient variance and alleviates sharpness. When utilized together, the spatio-temporal adaptation and the asymmetric components complement each other, effectively addressing the significant limitations of surrogate-gradient learning. They stabilize the optimization process and yield flatter solutions with improved generalization. The overall framework of $A^2SG$ is illustrated in Fig. 2.

## 4.1   RELATION BETWEEN VARIATION OF LOCAL GRADIENT AND FLATNESS OF LOSS LANDSCAPE

To analyze how the variability of local gradients affects the flatness of the loss landscape, we focus on the coefficient of variation (CV) of the local gradients. For notational simplicity, we consider a fully connected (FC) layer, but the principle naturally extends to other neural network layers and architectures. Let $W \in \mathbb{R}^{m \times n}$ denote the weight matrix of the FC layer. The gradient with respect to $W$, vectorized, is given by: $\mathbf{g} = \text{vec}\left(\frac{\partial L}{\partial W}\right) = \mathbf{a}_{\text{in}} \otimes \boldsymbol{\delta}$, where $\mathbf{a}_{\text{in}} \in \mathbb{R}^n$ is the input vector and

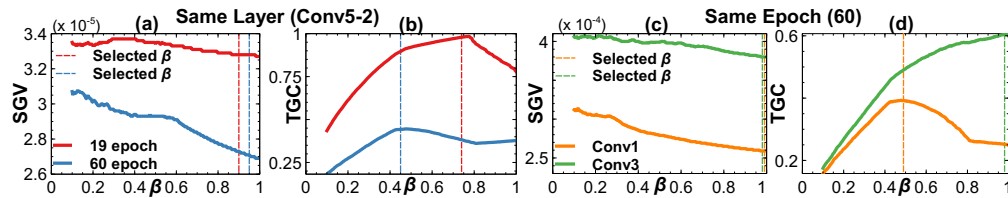

Figure 3: Graphs of SGV and TGC as a function of $\beta$. (a) and (b) represent SGV and TGC, at different epochs for the Conv5-2 layer, while (c) and (d) compare Conv1 and Conv3 at epoch 60. Dashed lines indicate the selected $\beta$ through the proposed adaptive method in each training case.

$\delta \in \mathbb{R}^m$ is the backpropagated error vector. The FIM is then defined as:

$$\mathbf{F} = \mathbb{E}[\mathbf{g}\mathbf{g}^\top] = \mathbb{E}\left[(\mathbf{a}_{\text{in}} \otimes \boldsymbol{\delta})(\mathbf{a}_{\text{in}} \otimes \boldsymbol{\delta})^\top\right]. \tag{5}$$

For analytical clarity, the $\delta$ can be decomposed into its mean and a zero-mean fluctuation term: $\boldsymbol{\delta} = \mu\mathbf{1} + \boldsymbol{\epsilon}$, where the $\mu$ os the mean of $\delta$, $\mathbf{1}$ is the all-ones vector and $\epsilon$ denotes a small perturbation. Substituting this into the definition of $\mathbf{F}$, we obtain:

$$\mathbf{F} = \mu^2 \mathbb{E}\left[(\mathbf{a}_{\text{in}} \otimes \mathbf{1})(\mathbf{a}_{\text{in}} \otimes \mathbf{1})^\top\right] + \mathbf{R} = \mathbf{F_0} + \mathbf{R}, \tag{6}$$

where $\mathbf{F_0}$ is a rank-1 matrix and the perturbation $\mathbf{R}$ is bounded as $||R||_2 \leq c\mu\text{CV}(\delta)$ for some constant $c$. This shows that as $\text{CV}(\delta)$ becomes smaller, the FIM approaches the rank-1 matrix $F_0$, indicating that the loss landscape has a dominant curvature direction. By matrix perturbation theory (Greenbaum et al., 2020), the largest eigenvalue of the FIM is bounded as:

$$\lambda_{\max}(F) \leq \mu^2 \lambda_{\max}\left(\mathbb{E}\left[(a_{\text{in}} \otimes \mathbf{1})(a_{\text{in}} \otimes \mathbf{1})^\top\right]\right) + c\mu^2\text{CV}(\delta), \tag{7}$$

where $\lambda_{\max}(\cdot)$ denotes the largest eigenvalue operator. Therefore, the largest eigenvalue grows linearly with $\text{CV}(\delta)$, and reducing the CV of local gradients directly leads to a flatter loss landscape.

## 4.2 SPATIO-TEMPORAL ADAPTIVE SURROGATE GRADIENTS (ST-ASG)

As discussed in the previous section, reducing local gradient variability alleviates the sharpness of the loss landscape. In addition, temporal gradient confusion can be mitigated by promoting alignment of local gradients across timesteps. To achieve this, we introduce two metrics: spatial gradient variation (SGV) and temporal gradient consistency (TGC), which guide spatial and temporal adaptation, respectively. SGV is defined as follows:

$$\text{SGV}^{(l)}[T] := \frac{\text{Var}(\boldsymbol{\delta}^l[T])}{\text{Mean}(|\boldsymbol{\delta}^l[T]|)}, \tag{8}$$

where $\delta^{(l)}[T]$ denotes the backpropagated error in layer $l$ at the last timestep $T$. To improve computational efficiency in practice, SGV employs the variance rather than the standard deviation in the denominator, differing from the conventional CV. On the other hand, TGC at timestep $t$ is defined as the cosine similarity between the local gradients at adjacent timesteps:

$$\text{TGC}^{(l)}[t] := \cos(\boldsymbol{\delta}^{(l)}[t], \boldsymbol{\delta}^{(l)}[t+1]), t \in [1, T-1]. \tag{9}$$

With these definitions, spatial adaptation is applied using SGV to suppress local gradient variation at the last timestep, where activations and gradients are relatively stable. In this case, the adaptation objective is to minimize SGV. Conversely, temporal adaptation is guided by TGC, which promotes alignment of local gradients between adjacent timesteps. In this context, the goal of adaptation is to maximize TGC. Since spatial adaptation is applied at the last timestep, it provides a stable reference direction for the temporal gradients. Temporal adaptation, applied to preceding timesteps ($t < T$), then aligns the local gradients with this reference. In this way, spatio-temporal adaptation is achieved by anchoring the global gradient trajectory to the stable direction obtained at the last timestep, while simultaneously enforcing temporal consistency across earlier timesteps.

To realize this spatio-temporal adaptation in practice, we propose to adjust the width ($\beta$) of the effective window. Our method is motivated by the observation that both SGV and TGC can be expressed as functions of $\beta$. As illustrated in Fig. 3, these functions vary unpredictably with the training dynamics: the same layer exhibits different functional shapes across epochs, and even within a single epoch, distinct patterns may emerge across layers. To robustly identify suitable values of $\beta$ under such variability, we employ a Bayesian search strategy. Implementation details of the adaptive method and the Bayesian optimization procedure are provided in Secs. A.7 and A.8, respectively.

### 4.3 ASYMMETRIC SURROGATE GRADIENTS

Symmetric functions, such as *TRI* and *BOX*, have been widely used as surrogate gradients in STBP, providing gradients based solely on the distance between a neuron's membrane potential and the threshold. However, they fail to account for neuronal dynamics such as integration and firing. Thus, the relative magnitude of the accumulated membrane potential is not effectively reflected in the training process. To solve this problem, we propose an asymmetric (*ASY*) surrogate, defined as

$$\frac{\partial s}{\partial u} = f(u, \beta) = \frac{1}{2\beta} \cdot (u - V_{\text{th}}) + h, \quad u \in [V_{\text{th}} - \beta, V_{\text{th}} + \beta]. \tag{10}$$

$h$ is empirically determined considering the gradient sparsity of each model. The proposed *ASY* function produces a larger gradient when the membrane potential is more highly accumulated, allowing the learning algorithm to capture each neuron's contribution based on its dynamic behavior.

This neuron's behavior-aware design also leads to a reduction in gradient variability. By concentrating gradient values in regions where membrane potential is high (and spike likelihood is greater), the *ASY* function avoids spreading gradients across irrelevant low-activity regions. This focused gradient allocation reduces unnecessary variability, resulting in more stable training.

This intuition is formalized in the following theoretical results.

**Theorem 1** (CV-Minimizing Symmetric Function under Area and Boundary Constraints). *Let* $f : [a, b] \to \mathbb{R}_{\geq 0}$ *be a function Let* $f_{\text{asy}}(u)$ *and* $f_{\text{sym}}(u)$ *be asymmetric and symmetric surrogate gradient functions defined over* $[a, b]$, *satisfying the boundary condition* $f(a) = f(b) = 0$, *non-negativity* $f(u) \geq 0$, *and area constraint* $\int_a^b f(u) \, du = c$, *where* $a = \theta - \beta$ *and* $b = \theta + \beta$. *Suppose the membrane potential* $u \sim \mathcal{N}(\mu, \sigma^2)$ *with* $\mu < a$, *so that* $p(u)$ *is decreasing on* $[a, b]$. *Then the unique function* $f^*$ *that minimizes the CV over all such admissible functions is the symmetric triangular function.*

*Proof.* Please refer to Thm. A1. □

Thm. 1 shows that, under area and boundary constraints, the symmetric function that minimizes gradient CV is a triangular function. This sets a lower bound of CV for symmetric functions under the given constraints. Based on this fact, we verify that asymmetric has a lower CV than symmetric.

**Theorem 2** (CV Comparison of Asymmetric and Symmetric Surrogates). *Let* $f_{\text{asy}}(u)$ *and* $f_{\text{sym}}(u)$ *be asymmetric and symmetric surrogate gradient functions defined over* $[a, b]$ *under the same constraints as in Thm. 1 Then, under a linear approximation of the Gaussian, where* $L = b - a$ *and* $\kappa = a - \mu$, *we have:*

$$\text{CV}_{\text{asy}} < \text{CV}_{\text{sym}} \quad \text{if } L\kappa > \sigma^2.$$

*Proof.* Please refer to Thm. A2. □

Thms. 1 and 2 demonstrate that the proposed asymmetric surrogate gradient, by incorporating membrane potential accumulation, achieves a lower CV than its symmetric counterparts. These theoretical findings indicate that the asymmetric surrogate gradient not only better reflects neuronal dynamics but also promotes more stable and efficient learning, thereby facilitating convergence to flatter minima. Experimental validations of Thms. 1 and 2 are provided in Fig. 5-(a), where *ASY* function consis-

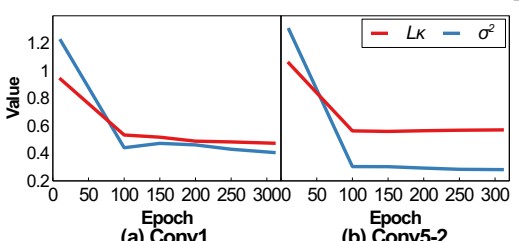

Figure 4: $L\kappa$ and $\sigma^2$ values across epochs. (a) and (b) correspond to Conv1 and Conv5-2, resepctively.

tently exhibits lower gradient variance than *TRI* function. Fig. 4 presents $L\kappa$ and $\sigma^2$ of Conv1 and Conv5-2 during training on VGG16 with CIFAR10. The graphs show that the condition $L\kappa > \sigma^2$ in Thm. 2 becomes satisfied across layers as training progresses. This confirms that our assumption is supported by the experimental results.

## 5 EXPERIMENTS

We evaluated the effectiveness of the proposed method on various datasets, including static image datasets such as CIFAR10, CIFAR100 (Krizhevsky et al., 2009), and ImageNet (Deng et al., 2009) as

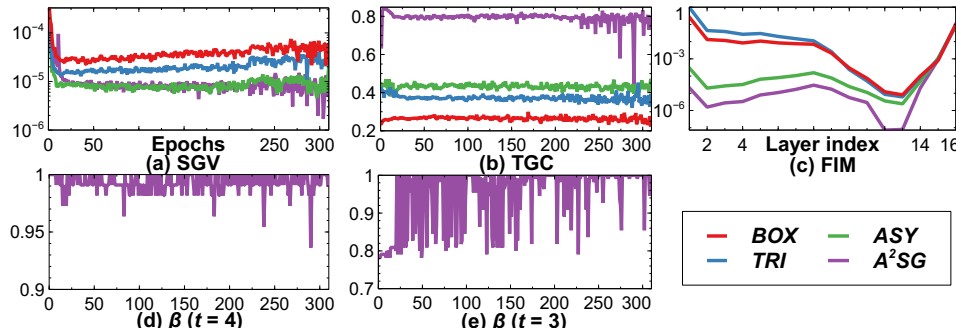

Figure 5: Comparison of SGV, TGC, FIM, and $\beta$ dynamics across different surrogate gradient functions at Conv5-2 layer. (a) SGV over epochs, (b) TGC over epochs, (c) FIM across layer indices, (d) $\beta$ dynamics at $t = 4$, and (e) $\beta$ dynamics at $t = 3$.

well as a neuromorphic dataset such as CIFAR10-DVS (Li et al., 2017). We conducted experiments on both CNN and Transformer models. To further show the versatility of our method, we also evaluated it on the ADE20K (Zhou et al., 2017) dataset for semantic segmentation. For more details about the experimental setup, please refer to Secs. A.9 and A.10.

## 5.1 EFFECT OF $A^2SG$ ON GRADIENT DYNAMICS AND FEATURE LEARNING

We analyze the effect of $A^2SG$ on gradient variation and temporal consistency. As shown in Fig. 5-(a) and (b), $A^2SG$ maintains low SGV and high TGC throughout training. Furthermore, in Fig. 5-(c), measurement of the maximum eigenvalue of the FIM for each layer reveals that $A^2SG$ consistently achieves the lowest eigenvalues throughout the network. These results corroborate our theoretical findings, demonstrating that lower CV directly leads to convergence toward flatter minima in the loss landscape. In addition, Fig. 5-(d) and (e) illustrate the adaptive selection of $\beta$ across training epochs, where $\beta$ is chosen to minimize SGV and maximize TGC, respectively. Additional analysis of the *ASY* function and the results for the Conv1 and Conv3 layers are provided in Secs. A.12 and A.13, respectively.

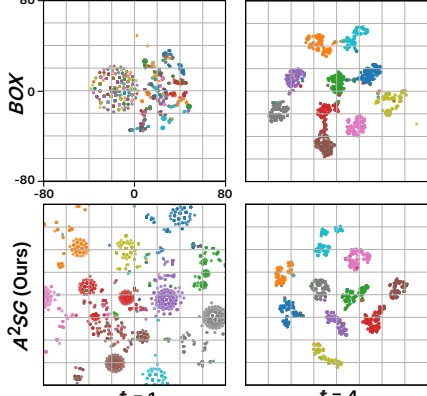

Figure 6: Comparison of t-SNE for VGG16 on CIFAR10 using *BOX* (top) and $A^2SG$ (bottom) at $t = 1$ and 4.

To further examine how improved gradient dynamics translate into feature representations, we perform a t-SNE visualization of the learned feature representations (Fig. 6). Models trained with $A^2SG$ exhibit class-separable features at early timestep $t = 1$ and by $t = 4$ the features of each class consolidate into compact and well-separated clusters. These properties indicate that gradients are effectively propagated to deeper layers and maintain coherent directions across timesteps, facilitating convergence to flat minima and improving training ability. In contrast, *BOX* exhibits significant class overlap and less clear separation. This demonstrates that $A^2SG$ enhances both training stability and generalization performance, as visualized by more robust and discriminative feature representations.

## 5.2 COMPARISON WITH OTHER ADAPTIVE SURROGATE GRADIENTS

Tab. 1 compares the accuracy of various adaptive surrogate gradient methods on CIFAR10 and CIFAR100 with ResNet18 and ResNet19. When applied to *BOX* and *TRI*, our ST-ASG consistently outperforms previous adaptive surrogate gradient methods on both datasets. This advantage arises from its spatio-temporal adaptation. The spatial policy reduces variability in local gradients, encouraging convergence toward flatter minima. The temporal policy aligns gradient directions across timesteps, stabilizing the global update. Furthermore, $A^2SG$ achieves the highest accuracy among all methods. On ResNet19, our approach outperforms LSG by $1.57\%$ on CIFAR10 ($96.74\%$ vs. $95.17\%$) and by $4.20\%$ on CIFAR100 ($81.05\%$ vs. $76.85\%$). These results demonstrate the strong generalization capability of our method over existing adaptive surrogate gradient approaches.

Table 1: Comparison with other adaptive surrogate gradient methods (*ARC*: Arctangent, ST-ASG: Our spatio-temporal adaptive surrogate gradient).

| Datasets | Architectures | Functions | Methods | Timesteps | Acc. (%) |
|---|---|---|---|---|---|
| CIFAR10 | ResNet18 | *ARC* | Dspike (Li et al., 2021) | 4 | 93.66±0.05 |
| | ResNet19 | *TRI* | CPNG (Lin et al., 2023) | 6 | 94.10±0.05 |
| | | *TRI* | ST-ASG | 4 | 96.41±0.16 |
| | | *BOX* | LSG (Lian et al., 2023) | 4 | 95.17±0.05 |
| | | *BOX* | ST-ASG | 4 | 96.01±0.05 |
| | | *ASY* | ***A²SG*** | **4** | **96.74±0.05** |
| CIFAR100 | ResNet18 | *ARC* | Dspike (Li et al., 2021) | 4 | 73.35±0.14 |
| | ResNet19 | *TRI* | CPNG (Lin et al., 2023) | 6 | 75.37±0.05 |
| | | *TRI* | ST-ASG | 4 | 80.46±0.06 |
| | | *BOX* | LSG (Lian et al., 2023) | 4 | 76.85±0.10 |
| | | *BOX* | ST-ASG | 4 | 78.60±0.16 |
| | | *ASY* | ***A²SG*** | **4** | **81.05±0.05** |

Table 2: Comparison with current state-of-the-art approaches on CIFAR10/100.

| Datasets | Architectures | Methods | Timesteps | Acc. (%) |
|---|---|---|---|---|
| CIFAR10 | VGG16 | IM (Guo et al., 2022) | 5 | 93.85 |
| | | RMP (Guo et al., 2023a) | 4 | 93.33 |
| | | MPBN (Guo et al., 2023b) | 4 | 94.44 |
| | | ***A²SG*** | **4** | **95.29±0.05** |
| | ResNet19 | IM (Guo et al., 2022) | 4 | 95.40 |
| | | RMP (Guo et al., 2023a) | 4 | 95.51 |
| | | TET (Deng et al., 2022) | 4 | 94.44 |
| | | TAB (Jiang et al., 2024) | 4 | 94.76 |
| | | ShortcutBP (Guo et al., 2024) | 2 | 95.36 |
| | | MPD-AGL (Jiang et al., 2025) | 2 | 96.18 |
| | | | 4 | 96.35 |
| | | | 6 | 96.54 |
| | | ***A²SG*** | **2** | **96.34±0.02** |
| | | | **4** | **96.74±0.05** |
| CIFAR100 | VGG16 | IM (Guo et al., 2022) | 5 | 70.18 |
| | | RMP (Guo et al., 2023a) | 4 | 72.55 |
| | | MPBN (Guo et al., 2023b) | 4 | 74.74 |
| | | ***A²SG*** | **4** | **75.21±0.08** |
| | ResNet19 | RMP (Guo et al., 2023a) | 4 | 78.28 |
| | | TET (Deng et al., 2022) | 4 | 74.47 |
| | | TAB (Jiang et al., 2024) | 4 | 76.81 |
| | | ShortcutBP (Guo et al., 2024) | 2 | 77.79 |
| | | MPD-AGL (Jiang et al., 2025) | 2 | 78.84 |
| | | | 4 | 79.72 |
| | | | 6 | 80.49 |
| | | ***A²SG*** | **2** | **79.18±0.01** |
| | | | **4** | **81.05±0.05** |

## 5.3 COMPARISON WITH STATE-OF-THE-ART METHODS

As reported in Tabs. 2 and 3, our method outperforms prior approaches on CIFAR10 and CIFAR100 in accuracy and achieves both higher accuracy and lower power consumption on ImageNet. In the case of E-SpikeFormer (Tab. 3), it adopted integer LIF neurons with multiple thresholds, which is distinct from the conventional SNNs with LIF neurons. To efficiently train on large-scale datasets, it uses integer values as a substitute for the temporal spike trains. Thus, this mechanism implicitly incorporates temporal dynamics, even when $T$=1. The model can be regarded as operating with an implicit time step equal to the maximum integer spike count $D$, rather than a single time step. Based on this perspective, we aligned the time step of the conventional LIF neurons with the integer activation value of I-LIF. For example, when $T \times D$ is 1x4 for I-LIF, we applied S-ASG to neurons with an activation value of four, corresponding to the last time step. We then sequentially applied T-ASG to neurons with activation values of three, two, and one. For the effective window ($\beta$), we set it to be centered on each threshold, as in LIF with a single threshold. For example, if $\text{th}_i$ is a threshold at integer $i$, the effective window of $\text{th}_i$ is set to $[\text{th}_i - \beta_i, \text{th}_i + \beta_i]$. From this state, we changed $\beta_i$ through our adaptive method.

Tab. 4 shows that our method maintains higher accuracy on CIFAR10-DVS with only four timesteps. In Tab. 5, our approach also attains higher mIoU and reduced power consumption on ADE20K,

Table 3: Comparison with Transformer-based SNNs on ImageNet. Following E-Spikeformer Yao et al. (2025), timesteps are denoted as $T \times D$, where $T$ is the number of timesteps and $D$ indicates the upper bound of integer activations.

| Architecture | Methods | Param (M) | Power (mJ) | Time Steps | Acc. (%) |
|---|---|---|---|---|---|
| | SpikFormer (Zhou et al., 2023) | 66.3 | 21.5 | $4 \times 1$ | 74.8 |
| | Meta-SpikeFormer (Yao et al., 2024) | 31.3 | 32.8 | $4 \times 1$ | 77.2 |
| Transformer | E-SpikeFormer(Yao et al., 2025) | 10.0 | 3.0 | $1 \times 4$ | 78.5 |
| | E-SpikeFormer(Yao et al., 2025) | 173.0 | 35.6 | $1 \times 4$ | 84.7 |
| | **E-SpikeFormer + $A^2SG$** | **10.0** | **2.78** | $1 \times 4$ | **78.61±0.01** |
| | **E-SpikeFormer + $A^2SG$** | **173.0** | **35.64** | $1 \times 4$ | **85.43** |

Table 4: Comparisons with other works on CIFAR10-DVS (* denotes our implementation).

| Datasets | Architectures | Methods | Timesteps | Acc. (%) |
|---|---|---|---|---|
| CIFAR10-DVS | VGGSNN | STBP-tdBN (Zheng et al., 2021)* | 4 | 81.30±1.00 |
| | | HSD (Zhong et al., 2024) | 5 | 81.10 |
| | | TMC (Yan et al., 2025) | 4 | 81.76 |
| | | $A^2SG$ | 4 | **82.36±0.01** |

Table 5: Performance of segmentation on ADE20K. These methods use the pre-trained models on ImageNet as the backbone, then add segmentation heads for fine-tuning.

| Dataset | Methods | Param (M) | Power (mJ) | Time Steps | MIoU. (%) |
|---|---|---|---|---|---|
| ADE20K | Meta-SpikeFormer (Yao et al., 2024) | 16.5 | 88.1 | $4 \times 1$ | 33.6 |
| | E-SpikeFormer (Yao et al., 2025) | 11.0 | 27.2 | $1 \times 4$ | 40.1 |
| | **E-SpikeFormer + $A^2SG$** | 11.0 | **25.2** | $1 \times 4$ | **40.94** |

demonstrating its effectiveness for segmentation as well as its energy efficiency. Moreover, Fig. A2 illustrates qualitative improvements in segmentation when applied to E-SpikeFormer. Notably, $A^2SG$ converges rapidly, achieving competitive accuracy with only two timesteps, comparable to other state-of-the-art methods that require four timesteps. Overall, these results demonstrate that by improving the design of surrogate gradients, our method provides a general and principled strategy for enhancing the training performance of deep SNNs across diverse architectures and tasks.

## 5.4 ABLATION STUDIES

Tab. 6 summarizes ablation results on CIFAR10 with VGG16, highlighting the contributions of the adaptive and asymmetric components. Incorporating the spatial adaptive surrogate gradient (S-ASG) improves accuracy and reduces spike count, while the temporal adaptive surrogate gradient (T-ASG) alone preserves accuracy with a slight increase in spikes. Their combination (ST-ASG) further stabilizes training, and adding the asymmetric surrogate ($A^2SG$) yields the highest accuracy (95.29%) with the lowest spike count, confirming the benefit of integrating all components. To estimate the computational overhead of

Table 6: Ablation study on CIFAR10/100 with VGG16, comparing spatial (S-ASG), temporal (T-ASG), spatio-temporal adaptation (ST-ASG), and ST-ASG with *ASY* (A²SG).

| | Methods | Acc. (%) | # of Spikes ($\times 10^3$) | Latency (sec/epoch) |
|---|---|---|---|---|
| **CIFAR10** | *BOX* (Baseline) | 94.84±0.05 | 94.6±1.0 | 74 (+0%) |
| | w/ S-ASG | 94.97±0.04 | 80.0±0.8 | 82 (+11%) |
| | w/ T-ASG | 94.94±0.05 | 98.7±1.8 | 83 (+12%) |
| | w/ ST-ASG | 94.98±0.03 | 93.6±2.3 | 85 (+15%) |
| | $A^2SG$ **(Ours)** | **95.29±0.04** | **84.9±1.8** | 85 (+15%) |
| **CIFAR100** | *BOX* (Baseline) | 74.24±0.08 | 100.3±1.2 | 74 (+0%) |
| | w/ S-ASG | 74.57±0.08 | 93.2±0.7 | 82 (+11%) |
| | w/ T-ASG | 74.54±0.06 | 104.9±2.2 | 83 (+12%) |
| | w/ ST-ASG | 74.73±0.07 | 100.2±0.2 | 85 (+15%) |
| | $A^2SG$ **(Ours)** | **75.21±0.08** | **100.2±0.4** | 85 (+15%) |

the proposed method, we measured the wall-clock time of one training epoch for each ablation case. As shown in the table, the proposed search method incurs a computational overhead of up to approximately 15% compared to the baseline.

## 5.5 NOISE ROBUSTNESS

In this section, we analyze the noise robustness of $A^2SG$. We experimented with deletion noise, removing a proportion of spikes from each layer, and report the results in Tab. A3.

Compared to the *BOX*, $A^2SG$ achieves higher accuracy under the deletion noise, while also exhibiting a smaller reduction in total spike counts. In this section, we analyze the noise robustness of $A^2SG$. We experimented with deletion noise, removing a proportion of spikes from each layer, and report the results in Tab. A3.

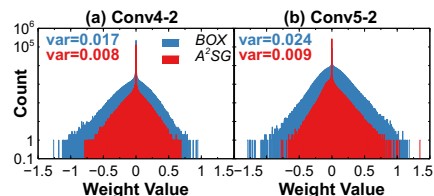

Fig. 7 illustrates the weight distributions, where $A^2SG$ shows lower variance that *BOX*. This distribution indicates reduced reliance on specific weights, thereby improving generalization and enhancing robustness to errors (Tsai et al., 2021).

Figure 7: Weight distribution of *BOX* and $A^2SG$. (a) and (b) are Conv4-2 and Conv5-2 layers

### 5.6 COMPATIBILITY

To validate the compatibility of our approach, we additionally applied $A^2SG$ to other neuron models and learning methods. Specifically, we evaluated it with PLIF neurons (Fang et al., 2021b) or RMP-loss (Guo et al., 2023a), and the corresponding results are reported in Tab. 7 and Tab. 8, respectively. Experimental results of PLIF or RMP-loss show improvements in both accuracy and efficiency, which is consistent with other experimental results. These experiments demonstrate that our method is compatible with various neuron models and training methods to improve both accuracy and efficiency.

Table 7: Comparison between LIF and PLIF on CIFAR10 with VGG16.

| Neurons | Methods | Acc. (%) | # of Spikes ($\times 10^3$) |
|---|---|---|---|
| LIF | Baseline | 94.84±0.05 | 94.6±1.0 |
| | $A^2SG$ (Ours) | **95.29±0.04** | **84.9±1.8** |
| PLIF Fang et al. (2021b) | Baseline | 94.99±0.03 | 91.6±7.4 |
| | $A^2SG$ (Ours) | **95.33±0.01** | **82.2±0.5** |

Table 8: Comparison between tdBN and RMP-loss on CIFAR100 with VGG16.

| Methods | Acc. (%) | # of Spikes ($\times 10^3$) |
|---|---|---|
| tdBN | 74.24±0.08 | 100.3±1.2 |
| RMP Guo et al. (2023a) | 74.39±0.07 | 102.8±2.9 |
| **tdBN + $A^2SG$** | **75.21±0.08** | **100.2±0.4** |
| **RMP + $A^2SG$** | **75.25±0.03** | **102.3±1.1** |

### 5.7 SENSITIVITY ANALYSIS

In this section, we conduct a sensitivity analysis of the hyperparameters of $A^2SG$. All results reported in the experiments use the following default configuration: $\beta$ update frequency of 1 epoch, Bayesian optimization parameters $(n_{obs}, n_{eval})$ set to (100, 150), and a search radius $\delta$ of 0.05. As shown in Tab. 9, increasing the update frequency to 50 or 100 epochs widens the interval between $\beta$ updates, which leads to accuracy drops within 0.2%. Similarly, when varying $(n_{obs}, n_{eval})$ to (10, 15) and (300, 450), the accuracy difference remained within 0.2%. Finally, we examined the

Table 9: Comparison with different $\beta$ update frequencies (epoch) and Bayesian optimization hyperparameter $(n_{obs}, n_{eval}, \delta)$ settings on CIFAR10 with VGG16.

| Epoch | $n_{obs}$ | $n_{eval}$ | $\delta$ | Acc. (%) |
|---|---|---|---|---|
| 1 | 100 | 150 | 0.05 | 95.29±0.04 |
| 50 | 100 | 150 | 0.05 | 95.10±0.02 |
| 100 | | | | 95.06±0.06 |
| 1 | 10 | 15 | 0.05 | 95.10±0.06 |
| | 300 | 450 | | 95.31±0.02 |
| 1 | 100 | 150 | 0.01 | 95.15±0.08 |
| | | | 0.10 | 95.24±0.03 |

sensitivity of $\delta$, which indicates the search width around $\beta$. The results obtained by varying it to 0.01 and 0.1 are reported in Tab. 9. Overall, the experimental observations indicate that $A^2SG$ demonstrates low sensitivity to hyperparameter variations.

## 6 CONCLUSION

In this work, we proposed $A^2SG$, a unified framework for training deep SNNs. By integrating spatio-temporal adaptation with a neuron-aware asymmetric design, $A^2SG$ reduces gradient variability, stabilizes optimization, and encourages convergence to flatter minima. Our theoretical analysis establishes the link between gradient variation and loss landscape curvature. Moreover, we prove that the asymmetric surrogate achieves lower variation than its symmetric counterparts. Extensive experiments across diverse SNN architectures and tasks demonstrate that $A^2SG$ consistently improves accuracy, robustness, and efficiency, highlighting surrogate gradient design as a key factor for reliable and scalable SNN training.

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

# A   APPENDIX

## A.1   LEAKY INTEGRATE-AND-FIRE (LIF) NEURON

The dynamics of a LIF neuron can be formulated as follows. First, the membrane potential is updated at each time step according to:

$$u_i^l[t] = \frac{1}{\tau}\left(v_i^l[t-1] + \sum_j w_{ij}^l s_j^{l-1}[t]\right), \tag{A1}$$

where $u$ denotes the membrane potential, $v$ the intermediate membrane state, $w$ the synaptic weights, and $s$ the input spike. Indices $i$ and $j$ represent the post- and pre-synaptic neurons, respectively, and $l$ refers to the layer index. The parameters $\tau$ and $t$ indicate the membrane time constant and discrete time step.

A spike is emitted when the membrane potential surpasses a predefined threshold:

$$s_i^l[t] = H\left(u_i^l[t] - V_{\text{th}}\right), \tag{A2}$$

where $H(\cdot)$ is the Heaviside step function and $V_{\text{th}}$ is the firing threshold.

After spike firing, the membrane potential is reset based on the intermediate state using the following mechanism:

$$v_i^l[t] = \left(u_i^l[t] - s_i^l[t]\right)s_i^l[t] + u_i^l[t]\left(1 - s_i^l[t]\right). \tag{A3}$$

## A.2   SPATIO-TEMPORAL BACKPROPAGATION (STBP)

SNNs require gradient propagation that considers not only spatial but also temporal variations. STBP is considered a suitable back-propagation method for SNNs as it incorporates both spatial and temporal components. The gradient of the loss $L$ with respect to the membrane potential $u_t^i[t]$ is defined as follows:

$$\frac{\partial L}{\partial u_i^l[t]} = \frac{\partial L}{\partial s_i^l[t]}\frac{\partial s_i^l[t]}{\partial u_i^l[t]} + \frac{\partial L}{\partial u_i^l[t+1]}\frac{\partial u_i^l[t+1]}{\partial u_i^l[t]}, \tag{A4}$$

here, $u[t]$ is the membrane potential at time $t$, and $s[t]$ is the spike counts at time $t$. The $l$ and $i$ indicate the layer and neuron index, respectively.

In addition, the gradient of the loss for the weight $w^l$ in $l$th layer is given by:

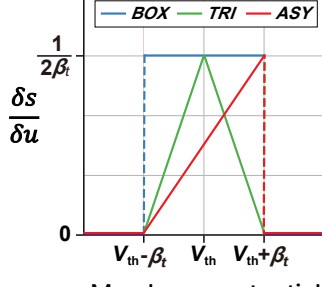

Figure A1: Visualization of surrogate gradient functions (*BOX*, *TRI*, *ASY*).

$$\frac{\partial L}{\partial w^l} = \sum_t \frac{\partial L}{\partial w^l[t]} = \sum_t \frac{\partial L}{\partial u^l[t]}s^{l-1}[t]. \tag{A5}$$

## A.3   SURROGATE GRADIENTS IN STBP

Surrogate gradient replaces the non-differentiable function $\frac{\partial s}{\partial u}$ with smooth approximated functions to enable gradient-based training. Representative surrogate gradient functions include the boxcar function and the triangle function, which are defined as follows:

$$\frac{\partial s}{\partial u} = \frac{1}{2\beta} \cdot 1(|u - V_{\text{th}}| < \beta), \tag{A6}$$

$$\frac{\partial s}{\partial u} = \frac{1}{\beta^2} \cdot \max(0, \beta - |u - V_{\text{th}}|). \tag{A7}$$

The above equations represent the *BOX* function and the *TRI* function, respectively, and $\beta$ is a parameter representing the effective window. These functions focus on mimicking the Dirac delta function, and their area remains constant regardless of the effective window. The function shapes are illustrated in Fig. A1

## A.4 FISHER INFORMATION MATRIX (FIM)

The FIM is defined as:

$$\mathbf{F}(\theta) = \mathbb{E}_{(x,y) \sim p(x,y;\theta)} \left[ \nabla_\theta \log p(y \mid x; \theta) \nabla_\theta \log p(y \mid x; \theta)^\top \right], \quad \text{(A8)}$$

where $\theta$ denotes the model parameters, $(x, y)$ is an input-output pair sampled from the data distribution $p(x, y; \theta)$, and $p(y \mid x; \theta)$ is the model's conditional probability. The operator $\nabla_\theta$ represents the gradient with respect to $\theta$. The FIM measures the variance of the gradient of the log-likelihood, reflecting the sensitivity of the model parameters to changes in the data distribution. In this context, the eigenvalues of the FIM quantify the curvature of the loss surface in various parameter directions.

## A.5 PROOF OF THE DERIVATIVES OF SYMMETRIC SURROGATE GRADIENTS

Let $f : [\theta - \beta, \theta + \beta] \to \mathbb{R}_{\geq 0}$ be a symmetric surrogate function, with a unique maximum at $u = \theta$, and

$$f(\theta - \beta) = f(\theta + \beta) = m \ (\geq 0), \qquad \int_{\theta-\beta}^{\theta+\beta} f(u) \, du = c > 0.$$

Set the *effective area above the floor* by

$$c_{\text{eff}} := \int_{\theta-\beta}^{\theta+\beta} \big( f(u) - m \big) \, du = c - 2\beta m.$$

We assume $c_{\text{eff}} > 0$ (otherwise $f \equiv m$ on the window and all derivatives vanish).

Define $g(u) := f(u) - m$. Then $g$ is symmetric, $g(\theta \pm \beta) = 0$, and $\int_{\theta-\beta}^{\theta+\beta} g(u) \, du = c_{\text{eff}}$. Let $M_f := f(\theta)$ and $M_g := g(\theta) = M_f - m$.

**Height.** By symmetry,

$$c_{\text{eff}} = 2 \int_\theta^{\theta+\beta} g(u) \, du \leq 2\beta \, M_g \quad \Rightarrow \quad M_g \geq \frac{c_{\text{eff}}}{2\beta},$$

hence

$$M_f = m + M_g \geq m + \frac{c_{\text{eff}}}{2\beta} = \frac{c}{2\beta}.$$

In particular, $\|f\|_\infty \geq M_f \geq c/(2\beta)$.

**First derivative.** On $[\theta, \theta + \beta]$, $g(\theta) = M_g$ and $g(\theta + \beta) = 0$. By the mean value theorem there exists $\xi$ with

$$|g'(\xi)| = \frac{M_g}{\beta}.$$

Since $g' = f'$, we obtain

$$\|f'\|_\infty \geq \frac{M_g}{\beta} \geq \frac{c_{\text{eff}}}{2\beta^2} = \frac{c - 2\beta m}{2\beta^2}.$$

**Conclusion.** With $c_{\text{eff}} = c - 2\beta m > 0$,

$$\|f\|_\infty \geq \frac{c}{2\beta} = \Omega(\beta^{-1}), \qquad \|f'\|_\infty \geq \frac{c - 2\beta m}{2\beta^2} = \Omega(\beta^{-2}).$$

Substituting into

$$\frac{\partial^2 L}{\partial w^2} = \frac{\partial^2 L}{\partial H^2}(H'(u)x)^2 + \frac{\partial L}{\partial H} H''(u) \, x^2 = \frac{\partial^2 L}{\partial H^2}(f(u)x)^2 + \frac{\partial L}{\partial H} f'(u) \, x^2, \quad \text{(A9)}$$

we obtain

$$\left| \frac{\partial^2 L}{\partial w^2} \right|_{SNN} = \Omega(\frac{x^2}{\beta^2}).$$

which shows sharper curvature than the $\mathcal{O}(x^2)$ scaling of smooth DNNs.

A.6   *TRI* INDUCES LARGER CURVATURE THAN *BOX* UNDER AREA NORMALIZATION

**Setup.**   Consider the surrogate gradients supported on $[\theta - \beta, \theta + \beta]$ with unit area. The boxcar (*BOX*) and triangular (*TRI*) surrogates are given by Eqs. A6 and A7. For a weight $w$ with input $x$ and pre-activation $u = wx$, the Hessian contribution is stated in Eq. A9.

**BOX properties.**   The *BOX* function has support length $2\beta$, constant height $1/(2\beta)$, and area 1. Therefore,

$$\|f_{\text{BOX}}\|_\infty = \tfrac{1}{2\beta}, \qquad \|f'_{\text{BOX}}\| = 0 \text{ almost everywhere on } (\theta - \beta, \theta + \beta).$$

**TRI properties.**   The *TRI* function is piecewise linear with the maximum at $u = \theta$ given by $f_{\text{TRI}}(\theta) = 1/\beta$, twice the peak of *BOX* under the same area constraint. The slopes are $\pm 1/\beta^2$, hence

$$\|f_{\text{TRI}}\|_\infty = \tfrac{1}{\beta}, \qquad \|f'_{\text{TRI}}\|_\infty = \tfrac{1}{\beta^2}.$$

**Comparison.**   Relative to *BOX*,

$$\|f_{\text{TRI}}\|_\infty = 2\,\|f_{\text{BOX}}\|_\infty, \qquad \|f'_{\text{TRI}}\|_\infty > \|f'_{\text{BOX}}\|_\infty.$$

Thus, *TRI* attains both higher peak and steeper slope.

**Implication for curvature.**   Substituting into Eq. A9, the first term scales with $f(u)^2$ and is therefore at least four times larger for *TRI* near $u = \theta$ compared to *BOX*. The second term involves $f'(u)$, which is strictly larger for *TRI* inside the window since $f'_{\text{BOX}} = 0$ almost everywhere. As a result, both contributions to the curvature are enhanced under *TRI*, leading to the following conclusion:

$$\left|\tfrac{\partial^2 L}{\partial w^2}\right|_{\text{TRI}} \;\geq\; 2 \cdot \left|\tfrac{\partial^2 L}{\partial w^2}\right|_{\text{BOX}}$$

A.7   ADAPTIVE SURROGATE GRADIENTS CONSIDERING SGV AND TGC

---

**Algorithm A1** Adaptive Surrogate Gradients Considering SGV and TGC

---

1: **Input:** iteration i, $\frac{\partial L}{\partial s[t]}$
2: **Output:** $\frac{\partial L}{\partial u[t]}$
3: **for** $l = L$ to 1 **do**
4:     **if** $i = 0$ **then**
5:         $\beta^l_{[t:0,\ldots,T]} \leftarrow \beta_{\text{init}}$
6:     **end if**
7:     **if** $((i \bmod i_{\text{update}}) = 0)$ and $(i \neq 0)$ **then**
8:         **for** $t = T$ to 1 **do**
9:             **if** $t = T$ **then**
10:                 $\beta^l_t \leftarrow \beta_{search}(SGV, \frac{\partial L}{\partial s[t]}, u^l[t])$
11:             **else**
12:                 $\beta^l_t \leftarrow \beta_{search}(TGC, g^l_{\text{cur}}[t], \frac{\partial L}{\partial s[t]}, u^l[t])$
13:             **end if**
14:         **end for**
15:     **else**
16:         **for** $t == T$ to 1 **do**
17:             $g^l_{\text{cur}}[t] \leftarrow f(u^l[t], \beta^l_t)$
18:         **end for**
19:     **end if**
20: **end for**

---

Alg. A1 outlines the adaptive surrogate gradient procedure that simultaneously considers both SGV and TGC. At the beginning of training, the effective window $\beta$ is initialized. During every update step, $\beta$ is adaptively adjusted according to the observed gradient distribution: SGV is used to calibrate the final timestep, while TGC is employed for earlier timesteps to maintain temporal consistency of error signals. When not in an update step, the algorithm computes the current layer gradient using the stored $\beta$. In this way, the method dynamically tunes $\beta$ across layers and timesteps.

## A.8 Beta Search via Bayesian Optimization

---

**Algorithm A2** Beta Search via Bayesian Optimization

---

1: **Input:** metric $M$, $g_{\text{ref}}$, $\frac{\partial L}{\partial s[t]}$, $u[t]$, $t$, $j$
2: **Output:** $\beta$
3: $\beta_{\min} \leftarrow \beta_{\min,\text{init}}$, $\beta_{\max} \leftarrow \beta_{\max,\text{init}}$
4: $\boldsymbol{\beta}_{\text{obs}} \leftarrow \text{Random}(\beta_{\min}, \beta_{\max}, n_{\text{obs}})$
5: **if** $t = T$ **then**
6: $\quad \boldsymbol{M} \leftarrow SGV(\boldsymbol{\beta}_{\text{obs}}, u[t], \frac{\partial L}{\partial s[t]})$
7: **else**
8: $\quad \boldsymbol{M} \leftarrow TGC(\boldsymbol{\beta}_{\text{obs}}, g_{ref}, u[t], \frac{\partial L}{\partial s[t]})$
9: **end if**
10: $f_{\text{best}} \leftarrow \max(\boldsymbol{M})$
11: $\beta_{best} \leftarrow \beta_{obs}[\text{argmax}(M)]$
12: $\beta_{min} \leftarrow \max(\beta_{best} - \delta, \beta_{\min})$
13: $\beta_{max} \leftarrow \min(\beta_{best} + \delta, \beta_{\max})$
14: $\boldsymbol{\beta}_{\text{eval}} \leftarrow \text{Uniform}(\beta_{\min}, \beta_{\max}, n_{\text{eval}})$
15: $\mu_s, \sigma_s \leftarrow \text{GP}(\boldsymbol{\beta}_{\text{obs}}, \boldsymbol{M}, \boldsymbol{\beta}_{\text{eval}})$
16: $\text{EI} \leftarrow \text{ExpectedImprovement}(\mu_s, \sigma_s, f_{\text{best}})$
17: $\beta^* \leftarrow \arg\max_\beta \text{EI}$
18: **if** $\beta^* < f_{\text{best}}$ **then**
$\quad \beta^* \leftarrow \beta_{\text{best}}$
19: **end if**
20: $\beta \leftarrow \beta^*$

---

Alg. A2 describes the adaptive search strategy for the effective window $\beta$. The method initializes a search range and samples candidates to evaluate the training metric. Depending on the current timestep $t$, either SGV or TGC is used to compute the evaluation metric. A Gaussian Process (GP) surrogate model is then fitted over the observed results, and the expected improvement (EI) criterion guides the selection of the next candidate $\beta$. The algorithm iteratively narrows down the search interval around the best candidate, ensuring efficient exploration while avoiding unstable regions. The final $\beta$ is set to the candidate with the highest improvement score.

## A.9 Computing Infrastructure

All experiments are performed on servers equipped with Intel(R) Xeon(R) Gold 6226R CPUs (2.90GHz, 520GB RAM) and NVIDIA RTX A6000 GPUs (8 units), running Ubuntu 20.04. Our implementation is based on CUDA 11.7, PyTorch 2.0.1 for ImageNet, Tensorflow/Keras 2.11.0 for CIFAR-10, CIFAR-100, and CIFAR10-DVS.

## A.10 Experimental setup

The input size of the model is set to 32x32 for CIFAR10/100, 224x224 for ImageNet, and 48x48 for CIFAR10-DVS. On CIFAR10/100, We trained each model for 310 epochs with the AdamW optimizer and a cosine decay learning rate scheduler with a 20-epoch warm-up. We set it to 200 epochs for CIFAR10-DVS. All models on CIFAR10 and CIFAR10-DVS were trained with an initial learning rate of $1 \times 10^{-5}$, a learning rate of $6 \times 10^{-3}$, and a weight decay of $2 \times 10^{-2}$. On CIFAR100, all models were trained with an initial learning rate of $1 \times 10^{-4}$, a learning rate of $5 \times 10^{-3}$, and a weight decay of $4 \times 10^{-2}$. Data augmentation was performed using a combination of CutMix Yun et al. (2019) and RandAugment Cubuk et al. (2020). RandAugment was configured with one augmentation per image, a magnitude of 1, a magnitude standard deviation of 0.4, and an application rate of 0.5. The batch size was set to 100 for CIFAR10/100. For ImageNet experiments, we employed the E-Spikeformer (Yao et al., 2025) model. A batch size of 360 was used for the 10M model, while a batch size of 100 was used for the 173M model. Both models were trained with a base learning rate of $6 \times 10^{-4}$, a minimum learning rate of $1 \times 10^{-6}$, and 5 warm-up epochs. For data augmentation, we applied Mixup (Zhang et al., 2018) with a weight of 0.8 and CutMix with a

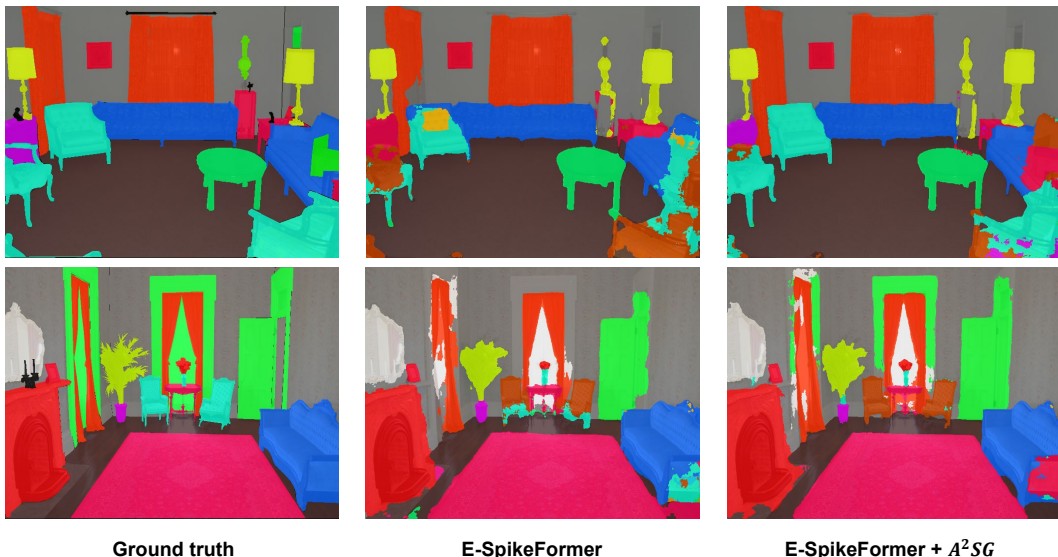

**Ground truth**          **E-SpikeFormer**          **E-SpikeFormer + $A^2SG$**

Figure A2: Segmentation results on the ADE20K dataset.

weight of 1.0. For all experiments, the effective window $\beta$ was initialized to 0.5, and the optimal $\beta$ during training was searched at the first iteration of every epoch.

### A.11 VISUALIZATION OF SEGMENTATION

Fig. A2 presents segmentation results on the ADE20K dataset. Compared to the E-spikeFormer, the proposed $A^2$SG improves boundary sharpness and object consistency.

### A.12 QUANTITATIVE ANALYSIS OF THE ASYMMETRIC SURROGATE GRADIENT

We provide additional analysis of the effect of *ASY* on VGG16 using CIFAR-10. As shown in Table A1, in the Conv1 layer, *ASY* substantially increases the proportion of silent neurons and reduces the total spike count (7.92k), outperforming both *BOX* (9.12k) and *TRI* (9.76k), while also yielding higher test accuracy.

Table A1: Distribution of spikes in the Conv1 layer for *BOX*, *TRI*, and *ASY*. Each value represents the percentage of samples with the corresponding spike counts.

| Spike Counts | BOX | TRI | ASY |
|---|---|---|---|
| 0 | 91.73% | 92.41% | 93.24% |
| 1 | 6.35% | 5.58% | 5.15% |
| 2 | 1.49% | 1.53% | 1.25% |
| 3 | 0.29% | 0.32% | 0.24% |
| 4 | 0.14% | 0.16% | 0.12% |
| **Num of Spikes** | 9.12k | 9.76k | 7.92k |

To further interpret this result, we analyze the membrane potential statistics. In SNNs, the weight distribution is mapped linearly to each neuron's membrane potential. Given an input spike count $M$, the membrane potential $u$ can be expressed as

$$u = \sum_i w_i x_i,$$

where $x_i$ and $w_i$ denote the presynaptic spikes and their corresponding weights, respectively. The mean and standard deviation of the membrane potential are $\mu_u = M\mu_w$ and $\sigma_u = \sqrt{M}\sigma_w$, where $\mu_w$ and $\sigma_w$ are the mean and standard deviation of the weight distribution. By the central limit theorem, the membrane potential can be approximated by $\mathcal{N}(\mu_u, \sigma_u^2)$. To quantify the distance between the mean membrane potential and the firing threshold $V_{\text{th}}$, we define the relative membrane distance (RMD) as

$$\text{RMD} := \frac{V_{\text{th}} - \mu_u}{\sigma_u}. \tag{A10}$$

A larger RMD indicates that the membrane potential lies farther below the threshold, thus reducing the probability of spiking.

Table A2: $\mu_w$, $\sigma_w$, and RMD depending on the surrogate gradient (SG) functions in Conv1.

| SG func. | $\mu_w$ ($\times 10^{-3}$) | $\sigma_w$ | RMD | # of spikes ($\times 10^3$) |
|---|---|---|---|---|
| *BOX* | -8.122 | 0.1408 | 7.159 | 9.12 |
| *TRI* | -8.205 | 0.1394 | 7.231 | 9.76 |
| *ASY* | -7.002 | 0.1361 | 7.398 | 7.92 |

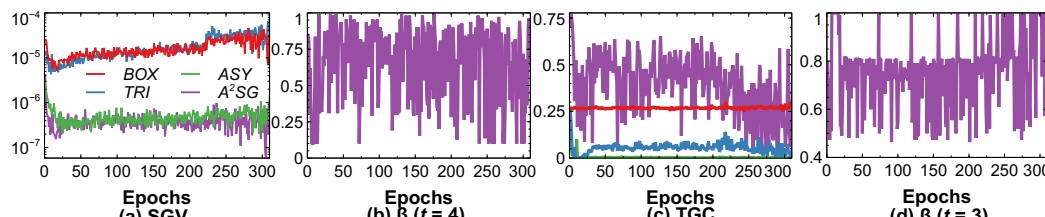

**(a) SGV**    **(b) β ($t = 4$)**    **(c) TGC**    **(d) β ($t = 3$)**

Figure A3: Comparison of SGV, TGC, and $\beta$ dynamics across different surrogate gradient functions at Conv1 layer. (a) SGV over epochs, (b) $\beta$ dynamics at $t = 4$, (c) TGC over epochs, (d) $\beta$ dynamics at $t = 3$.

As summarized in Table A2, *ASY* achieves the highest RMD and the lowest spike count, confirming its effectiveness in suppressing redundant spikes and improving energy efficiency.

### A.13 COMPARISON OF SGV, TGC, AND $\beta$ DYNAMICS ON OTHER LAYERS

Fig. A3 and A4 illustrate SGV, TGC, and $\beta$ for Conv1 and Conv3, respectively. Similar to the results in Conv5-2, $A^2SG$ achieves the lowest SGV and the highest TGC. Notably, while *ASY* exhibited relatively high TGC in Conv5-2, it shows much lower TGC in the earlier layers. In contrast, $A^2SG$ consistently maintains high TGC across all layers by applying the adaptive surrogate strategy to *ASY*. Furthermore, in Figs. A3 and A4-(b), (d), it can be observed that $\beta$ is adjusted in a manner that improves both SGV and TGC.

### A.14 THEORETICAL ANALYSIS

**Theorem A1** (CV-Minimizing Symmetric Function under Area and Boundary Constraints)**.** *Let $f : [a, b] \to \mathbb{R}_{\geq 0}$ be a function satisfying, where $a = \theta - \beta$ and $b = \theta + \beta$:*

- *(**Symmetry**): $f(u) = f(a + b - u)$ for all $u \in [a, b]$*

- *(**Boundary condition**): $f(a) = f(b) \geq 0$*

- *(**Nonnegativity**): $f(u) \geq 0$*

- *(**Area constraint**): $\int_a^b f(u)\, du = c > 0$*

*Let the weight function be given by the Gaussian density*

$$p(u) := \frac{1}{\sqrt{2\pi}\sigma} \exp\left(-\frac{(u - \mu)^2}{2\sigma^2}\right), \quad \text{with } \mu < a,$$

*so that $p(u)$ is strictly decreasing and convex on $[a, b]$.*

*Then the unique function $f^*$ that minimizes the CV*

$$\mathbf{CV_{sim}}[f] := \frac{\sqrt{\int_a^b f(u)^2 p(u)\, du - \left(\int_a^b f(u)p(u)\, du\right)^2}}{\int_a^b f(u)p(u)\, du},$$

*over all such admissible functions is the symmetric triangular function*

$$f^*(u) := \frac{4c}{(b - a)^2} \cdot \min(u - a,\ b - u).$$

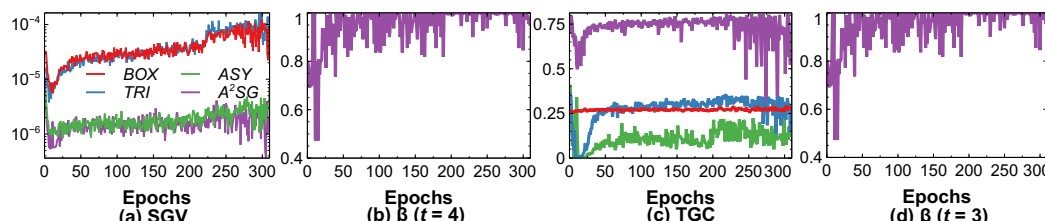

Figure A4: Comparison of SGV, TGC, and $\beta$ dynamics across different surrogate gradient functions at Conv3 layer. (a) SGV over epochs, (b) $\beta$ dynamics at $t = 4$, (c) TGC over epochs, (d) $\beta$ dynamics at $t = 3$.

*Proof.* Let $m = \frac{a+b}{2}$ be the midpoint of the interval. Due to the symmetry condition, any admissible function $f$ is uniquely determined by its restriction $g$ to $[a, m]$, with the reconstruction:

$$f(u) = \begin{cases} g(u), & u \in [a, m], \\ g(a + b - u), & u \in [m, b]. \end{cases}$$

Since $p(u)$ is strictly decreasing and convex on $[a, b]$, it is also decreasing on $[a, m]$. Over this domain, define the following Rayleigh-type quotient:

$$R[g] := \frac{\int_a^m g(u)^2 p(u)\, du + \int_m^b g(a + b - u)^2 p(u)\, du}{\left( \int_a^m g(u) p(u)\, du + \int_m^b g(a + b - u) p(u)\, du \right)^2}.$$

By change of variable $u' = a + b - u$, and using symmetry of $f$, we have:

$$R[g] = \frac{2 \int_a^m g(u)^2 p(u)\, du}{\left( 2 \int_a^m g(u) p(u)\, du \right)^2} = \frac{\int_a^m g(u)^2 p(u)\, du}{\left( \int_a^m g(u) p(u)\, du \right)^2}.$$

This Rayleigh quotient is minimized when $g(u)$ is proportional to a linear function increasing from $a$ to $m$. That is:

$$g^*(u) = \alpha(u - a), \quad u \in [a, m],$$

with boundary condition $g(a) = 0$. Extending symmetrically gives:

$$f^*(u) = \alpha \cdot \min(u - a, b - u).$$

To satisfy the area constraint:

$$\int_a^b f^*(u)\, du = \int_a^m \alpha(u - a)\, du + \int_m^b \alpha(b - u)\, du = \alpha \cdot \frac{(b - a)^2}{4} = c,$$

which yields:

$$\alpha = \frac{4c}{(b - a)^2}.$$

Thus, the minimizing function is:

$$f^*(u) = \frac{4c}{(b - a)^2} \cdot \min(u - a, b - u),$$

which is the symmetric triangular function. $\qquad\square$

**Theorem A2** (CV Comparison of Asymmetric and Symmetric Surrogates). *Let $f_{\mathrm{asy}}(u)$ and $f_{\mathrm{sym}}(u)$ be asymmetric and symmetric surrogate gradient functions defined over $[a, b]$, satisfying the boundary condition $f(a) = f(b) = 0$, nonnegativity $f(u) \geq 0$, and area constraint $\int_a^b f(u)\, du = c$, where $a = \theta - \beta$ and $b = \theta + \beta$. Suppose the membrane potential $u \sim \mathcal{N}(\mu, \sigma^2)$ with $\mu < a$, so that $p(u)$ is strictly decreasing on $[a, b]$. Then, under a linear approximation of the Gaussian, we have:*

$$\mathrm{CV}_{\mathrm{asy}} < \mathrm{CV}_{\mathrm{sym}} \quad \textit{if } L\kappa > \sigma^2.$$

*Proof.* We define $L := b - a$ and approximate the Gaussian by a first-order Taylor expansion around $u = a$:

$$A := p(a) = \frac{1}{\sqrt{2\pi}\sigma} \exp\left(-\frac{(a-\mu)^2}{2\sigma^2}\right),$$

$$B := -p'(a) = \frac{a-\mu}{\sigma^2} \cdot A.$$

Let $\kappa := a - \mu$, so that:

$$A = \frac{1}{\sqrt{2\pi}\sigma} e^{-\frac{\kappa^2}{2\sigma^2}}, \quad B = \frac{\kappa}{\sigma^2} A.$$

Under the area constraint, define the surrogates:

$$f_{\text{asy}}(u) = \frac{2c}{L} \cdot \frac{u-a}{L},$$

$$\text{and} \quad f_{\text{sym\_tri}}(u) = \begin{cases} \frac{2c}{L} \cdot \frac{u-a}{L/2}, & u \le \theta, \\ \frac{2c}{L} \cdot \frac{b-u}{L/2}, & u > \theta, \end{cases} \quad \text{where } \theta = \frac{a+b}{2}.$$

**Asymmetric surrogate:**

$$f_{\text{asy}}(u) = \frac{2c}{L^2}(u-a)$$

*Expectation:*

$$\mathbb{E}[f_{\text{asy}}] = \int_a^b f_{\text{asy}}(u) \cdot p(u)\, du$$

$$= \frac{2c}{L^2} \int_a^b (u-a)(A - B(u-a))\, du$$

$$= \frac{2c}{L^2} \int_0^L x(A - Bx)\, dx$$

$$= \frac{2c}{L^2}\left(A \cdot \frac{L^2}{2} - B \cdot \frac{L^3}{3}\right)$$

$$= c\left(A - \frac{2}{3}BL\right)$$

*Second moment:*

$$\mathbb{E}[f_{\text{asy}}^2] = \left(\frac{2c}{L^2}\right)^2 \int_0^L x^2(A - Bx)\, dx$$

$$= \frac{4c^2}{L^4}\left(A \cdot \frac{L^3}{3} - B \cdot \frac{L^4}{4}\right)$$

$$= \frac{4c^2}{L}\left(\frac{A}{3} - \frac{BL}{4}\right)$$

*Variance:*

$$\text{Var}[f_{\text{asy}}] = \mathbb{E}[f_{\text{asy}}^2] - \mathbb{E}[f_{\text{asy}}]^2$$

$$= \frac{4c^2}{L}\left(\frac{A}{3} - \frac{BL}{4}\right) - c^2\left(A - \frac{2}{3}BL\right)^2$$

**Symmetric surrogate (triangle function):**

$$f_{\text{sym\_tri}}(u) = \begin{cases} \frac{4c}{L^2}(u-a), & u \in [a, \theta], \\ \frac{4c}{L^2}(b-u), & u \in [\theta, b] \end{cases}, \quad \theta = \frac{a+b}{2}$$

*Expectation:*

$$\mathbb{E}[f_{\mathrm{sym\_tri}}] = \frac{4c}{L^2}\left[\int_a^\theta (u-a)p(u)\,du + \int_\theta^b (b-u)p(u)\,du\right]$$

$$= \frac{8c}{L^2}\int_0^{L/2} x(A-Bx)\,dx$$

$$= \frac{8c}{L^2}\left(A\cdot\frac{(L/2)^2}{2} - B\cdot\frac{(L/2)^3}{3}\right)$$

$$= c\left(A - \frac{1}{3}BL\right)$$

*Second moment:*

$$\mathbb{E}[f_{\mathrm{sym\_tri}}^2] = \frac{8c^2}{L^4}\int_0^{L/2} x^2(A-Bx)\,dx\cdot 2$$

$$= \frac{16c^2}{L^4}\left(A\cdot\frac{(L/2)^3}{3} - B\cdot\frac{(L/2)^4}{4}\right)$$

$$= \frac{2c^2}{L}\left(\frac{A}{3} - \frac{BL}{8}\right)$$

*Variance:*

$$\mathrm{Var}[f_{\mathrm{sym\_tri}}] = \frac{2c^2}{L}\left(\frac{A}{3} - \frac{BL}{8}\right) - c^2\left(A - \frac{1}{3}BL\right)^2$$

**Coefficient Ratio:**

$$r := \frac{\mathbb{E}[f_{\mathrm{asy}}]}{\mathbb{E}[f_{\mathrm{sym\_tri}}]} = \frac{A - \frac{2}{3}BL}{A - \frac{1}{3}BL} = \frac{3A - 2BL}{3A - BL}$$

$$\eta := \frac{\mathrm{Var}[f_{\mathrm{asy}}]}{\mathrm{Var}[f_{\mathrm{sym\_tri}}]} = \frac{\frac{4c^2}{L}\left(\frac{A}{3} - \frac{BL}{4}\right) - c^2\left(A - \frac{2}{3}BL\right)^2}{\frac{2c^2}{L}\left(\frac{A}{3} - \frac{BL}{8}\right) - c^2\left(A - \frac{1}{3}BL\right)^2}$$

Then the squared CV ratio becomes:

$$\left(\frac{\mathrm{CV_{asy}}}{\mathrm{CV_{sym\_tri}}}\right)^2 = \frac{\eta}{r^2}$$

$$= \frac{3(2A - BL)^2\cdot\left[12A - 9BL + L(3A - 2BL)^2\right]}{(3A - 2BL)^2\cdot\left[16A - 8BL + 3L(2A - BL)^2\right]},$$

which simplifies under substitution to:

$$\frac{\eta}{r^2} \sim \frac{3}{4}\cdot\left(\frac{L\kappa - 2\sigma^2}{2L\kappa - 3\sigma^2}\right)^2$$

and thus yields the condition $\mathrm{CV_{asy}} < \mathrm{CV_{sym\_tri}}$ if $L\kappa > \sigma^2$. Therefore, according to the Theorem 1, $\mathrm{CV_{asy}} < \mathrm{CV_{sym}}$ if $L\kappa > \sigma^2$.

$\square$

### A.15 Extension of Theorem 2 with $n$-segment piecewise-linear approximation

In this subsection, we show that the conclusion of Theorem 2 is stable when the Gaussian weight function on the effective window is approximated by an $n$-segment piecewise-linear model instead of a single linear segment. We use the same notation as in Theorem 2 and Theorem A2: the effective window is $I = [a, b] = [\theta - \beta, \theta + \beta]$ with width $L = b - a$, the membrane potential satisfies $u \sim \mathcal{N}(\mu, \sigma^2)$ with $\mu < a$, and $p(u)$ denotes the Gaussian weight function. We also define $\delta := a - \mu > 0$.

**One-segment and $n$-segment PWLA models.** We now specify the weight-function models on $I$.

- **One-segment linear model $\tilde{p}_1$.** Following Theorem 2 (and Theorem A2), we approximate $p(u)$ on $I$ by its first-order Taylor expansion at the left boundary $u = a$:

$$\tilde{p}_1(u) := p(a) + p'(a)\,(u - a), \qquad u \in I.$$

- **$n$-segment piecewise-linear model $\tilde{p}_n$.** For $n \in \mathbb{N}$, we divide $I$ into $n$ equal sub-intervals of length $\Delta := L/n$ with grid points $u_k := a + k\Delta$ for $k = 0, \ldots, n$. The $n$-segment piecewise-linear approximation of $p$ is defined by

$$\tilde{p}_n(u) := p(u_k) + p'(u_k)\,(u - u_k) \quad \text{for } u \in [u_k, u_{k+1}),\ k = 0, \ldots, n - 1. \qquad \text{(A11)}$$

We denote $R_1^2 := R(\tilde{p}_1)^2$ and $R_n^2 := R(\tilde{p}_n)^2$.

We denote by $\mathcal{H}$ the class of weight functions considered in this paper, namely the Gaussian weight $p(u)$ and its PWLA approximations $\tilde{p}_1, \tilde{p}_n$ on the effective window $I = [\theta - \beta, \theta + \beta]$ We refer to elements of $\mathcal{H}$ as admissible weight functions.

**Lemma 1** (Bound on the variation of $R(h)^2$). *Let $I = [a, b]$ be the effective window. For any nonnegative weight function $h : I \to \mathbb{R}_{\geq 0}$, define*

$$M_1(h) = \int_a^b f_{\mathrm{asy}}(u)\,h(u)\,du, \qquad\qquad M_2(h) = \int_a^b f_{\mathrm{asy}}(u)^2\,h(u)\,du,$$

$$J_1(h) = \int_a^b f_{\mathrm{sym}}(u)\,h(u)\,du, \qquad\qquad J_2(h) = \int_a^b f_{\mathrm{sym}}(u)^2\,h(u)\,du.$$

*Based on these, we define*

$$\mathrm{CV}_{\mathrm{asy}}(h)^2 := \frac{M_2(h) - M_1(h)^2}{M_1(h)^2},$$

$$\mathrm{CV}_{\mathrm{sym}}(h)^2 := \frac{J_2(h) - J_1(h)^2}{J_1(h)^2},$$

$$R(h)^2 := \left(\frac{\mathrm{CV}_{\mathrm{asy}}(h)}{\mathrm{CV}_{\mathrm{sym}}(h)}\right)^2.$$

*Assume that there exist constants $m_{\min} > 0$ and $v_{\min} > 0$ such that, for all admissible weight functions $h$ in the class considered in this paper,*

$$M_1(h),\ J_1(h) \geq m_{\min}, \qquad M_2(h) - M_1(h)^2,\ J_2(h) - J_1(h)^2 \geq v_{\min}. \qquad \text{(A12)}$$

*Then there exists a constant $K > 0$ such that, for any admissible $h, \tilde{h}$,*

$$|R(h)^2 - R(\tilde{h})^2| \ \leq \ K\,\|h - \tilde{h}\|_\infty, \qquad\qquad \text{(A13)}$$

*where $\|h - \tilde{h}\|_\infty := \sup_{u \in I} |h(u) - \tilde{h}(u)|$.*

*Proof.* We denote $R(h)^2$ as
$$R(h)^2 = \Phi\big(v(h)\big),$$
where
$$v(h) := \big(M_1(h), M_2(h), J_1(h), J_2(h)\big) \in \mathbb{R}^4,$$
and
$$\Phi(m_1, m_2, j_1, j_2) := \frac{j_1^2}{m_1^2} \cdot \frac{m_2 - m_1^2}{j_2 - j_1^2}.$$

**Step 1: Sensitivity of $v(h)$ with respect to $h$.** Let $L := b - a$ and let $h, \tilde{h}$ be two admissible weight functions. Set $\Delta h(u) := h(u) - \tilde{h}(u)$. Since $f_{\mathrm{asy}}$ and $f_{\mathrm{sym}}$ are fixed surrogate functions on the effective window $I$, they are bounded:

$$C_1 := \sup_{u \in I}\big(|f_{\mathrm{asy}}(u)|,\ |f_{\mathrm{sym}}(u)|\big) < \infty, \quad C_2 := \sup_{u \in I}\big(f_{\mathrm{asy}}(u)^2,\ f_{\mathrm{sym}}(u)^2\big) < \infty.$$

We first bound the change in $M_1$:

$$M_1(h) - M_1(\tilde{h}) = \int_a^b f_{\mathrm{asy}}(u)\, h(u)\, du - \int_a^b f_{\mathrm{asy}}(u)\, \tilde{h}(u)\, du$$

$$= \int_a^b f_{\mathrm{asy}}(u)\, \Delta h(u)\, du.$$

Using the triangle inequality and the definition of the sup norm,

$$|M_1(h) - M_1(\tilde{h})| \le \int_a^b |f_{\mathrm{asy}}(u)|\, |\Delta h(u)|\, du$$

$$\le \left(\sup_{u \in I} |f_{\mathrm{asy}}(u)|\right) \int_a^b |\Delta h(u)|\, du$$

$$\le C_1 \cdot L \cdot \sup_{u \in I} |\Delta h(u)|$$

$$= C_1 L \,\|h - \tilde{h}\|_\infty.$$

Similarly, for $M_2$ we have

$$M_2(h) - M_2(\tilde{h}) = \int_a^b f_{\mathrm{asy}}(u)^2\, \Delta h(u)\, du,$$

and hence

$$|M_2(h) - M_2(\tilde{h})| \le C_2 L \,\|h - \tilde{h}\|_\infty.$$

The same argument with $f_{\mathrm{sym}}$ and $f_{\mathrm{sym}}^2$ yields

$$|J_1(h) - J_1(\tilde{h})| \le C_1 L \,\|h - \tilde{h}\|_\infty, \qquad |J_2(h) - J_2(\tilde{h})| \le C_2 L \,\|h - \tilde{h}\|_\infty.$$

Therefore, there exists a constant $C_v > 0$ ($C_v := L \max\{C_1, C_2\}$) such that

$$\|v(h) - v(\tilde{h})\| \;\le\; C_v \,\|h - \tilde{h}\|_\infty, \tag{A14}$$

where $\|\cdot\|$ denotes the Euclidean norm on $\mathbb{R}^4$.

**Step 2: Sensitivity of $\Phi(v)$ with respect to $v$.** By the assumptions in equation A12, for admissible weight functions $h$, we have

$$M_1(h), J_1(h) \ge m_{\min}, \qquad M_2(h) - M_1(h)^2,\ J_2(h) - J_1(h)^2 \ge v_{\min}.$$

Moreover, since the surrogates and weight functions are bounded on the finite interval $I$, the integrals $M_1(h), M_2(h), J_1(h), J_2(h)$ are bounded. Hence all vectors $v(h)$ lie in a set $K \subset \mathbb{R}^4$ on which:

- the denominators $m_1^2$ and $j_2 - j_1^2$ in the definition of $\Phi(m_1, m_2, j_1, j_2)$ are bounded away from zero by $m_{\min}^2$ and $v_{\min}$, and

- $\Phi$ is continuously differentiable.

In particular, the gradient is bounded on $K$:

$$C_\Phi := \sup_{v \in K} \|\nabla \Phi(v)\| < \infty.$$

Now, for any $v, \tilde{v} \in K$, consider the line segment $\gamma(t) := v + t(\tilde{v} - v)$, $t \in [0, 1]$, and define $\psi(t) := \Phi(\gamma(t))$. By the chain rule,

$$\psi'(t) = \nabla \Phi(\gamma(t)) \cdot (\tilde{v} - v),$$

so that

$$|\psi'(t)| \le \|\nabla \Phi(\gamma(t))\|\, \|\tilde{v} - v\| \le C_\Phi \,\|\tilde{v} - v\| \quad \text{for all } t \in [0, 1].$$

Integrating from 0 to 1, we obtain

$$|\Phi(\tilde{v}) - \Phi(v)| = |\psi(1) - \psi(0)|$$

$$= \left| \int_0^1 \psi'(t)\, dt \right|$$

$$\leq \int_0^1 |\psi'(t)|\, dt$$

$$\leq \int_0^1 C_\Phi \left\| \tilde{v} - v \right\| dt$$

$$= C_\Phi \left\| \tilde{v} - v \right\|.$$

Therefore

$$|\Phi(\tilde{v}) - \Phi(v)| \;\leq\; C_\Phi \left\| \tilde{v} - v \right\|. \tag{A15}$$

**Step 3: Combining the two bounds.** Finally, for $h, \tilde{h}$ we have

$$|R(h)^2 - R(\tilde{h})^2| = |\Phi(v(h)) - \Phi(v(\tilde{h}))|.$$

Applying equation A15 with $v = v(h)$ and $\tilde{v} = v(\tilde{h})$, and then equation A14, we obtain

$$|R(h)^2 - R(\tilde{h})^2| \leq C_\Phi \left\| v(h) - v(\tilde{h}) \right\| \leq C_\Phi C_v \left\| h - \tilde{h} \right\|_\infty,$$

where $K = C_\Phi C_v$. $\qquad\square$

**Lemma 2** (Approximation error of $\tilde{p}_1$ and $\tilde{p}_n$)**.** *Let $I = [a, b]$ be the effective window with length $L = b - a$. Assume that $p$ is twice continuously differentiable on $I$, and define*

$$C_{p''} := \sup_{u \in I} |p''(u)| < \infty.$$

*Then, for all $u \in I$,*

$$|p(u) - \tilde{p}_1(u)| \;\leq\; \tfrac{1}{2} C_{p''} L^2, \qquad |p(u) - \tilde{p}_n(u)| \;\leq\; \tfrac{1}{2} C_{p''} \frac{L^2}{n^2},$$

*and hence*

$$\|\tilde{p}_1 - \tilde{p}_n\|_\infty \;\leq\; \tfrac{1}{2} C_{p''} L^2 \left( 1 + \tfrac{1}{n^2} \right). \tag{A16}$$

**Error of the one-segment linear approximation $\tilde{p}_1$.** Let $I = [a, b]$ and $L = b - a$. The one-segment linear model $\tilde{p}_1$ is defined as the first-order Taylor polynomial of $p$ at $u = a$:

$$\tilde{p}_1(u) := p(a) + p'(a)(u - a), \qquad u \in I.$$

By Taylor's theorem with Lagrange remainder at $u = a$, for each $u \in I$ there exists a point $\xi_u$ on the segment between $a$ and $u$ such that

$$p(u) = p(a) + p'(a)(u - a) + \frac{1}{2} p''(\xi_u)(u - a)^2.$$

Hence

$$p(u) - \tilde{p}_1(u) = \frac{1}{2} p''(\xi_u)(u - a)^2.$$

Taking absolute values and using the definition of $C_{p''}$,

$$|p(u) - \tilde{p}_1(u)| = \frac{1}{2} |p''(\xi_u)| \, |u - a|^2 \leq \frac{1}{2} C_{p''} \, |u - a|^2.$$

Since $u \in [a, b]$ implies $|u - a| \leq L$, we obtain

$$|p(u) - \tilde{p}_1(u)| \leq \frac{1}{2} C_{p''} L^2 \qquad \text{for all } u \in I.$$

**Error of the $n$-segment PWLA approximation $\tilde{p}_n$.** Partition $I = [a, b]$ into $n$ equal sub-intervals of length $\Delta := L/n$, and let $u_k := a + k\Delta$ for $k = 0, \ldots, n$. On each sub-interval $[u_k, u_{k+1}]$, the $n$-segment PWLA model is

$$\tilde{p}_n(u) := p(u_k) + p'(u_k)(u - u_k), \qquad u \in [u_k, u_{k+1}].$$

Applying Taylor's theorem at $u_k$ for $u \in [u_k, u_{k+1}]$, there exists $\xi_{k,u} \in [u_k, u_{k+1}]$ such that

$$p(u) = p(u_k) + p'(u_k)(u - u_k) + \frac{1}{2}p''(\xi_{k,u})(u - u_k)^2.$$

Therefore

$$p(u) - \tilde{p}_n(u) = \frac{1}{2}p''(\xi_{k,u})(u - u_k)^2,$$

and thus

$$|p(u) - \tilde{p}_n(u)| = \frac{1}{2}|p''(\xi_{k,u})|\,|u - u_k|^2 \leq \frac{1}{2}C_{p''}|u - u_k|^2.$$

Since $u \in [u_k, u_{k+1}]$ implies $|u - u_k| \leq \Delta = L/n$, we obtain

$$|p(u) - \tilde{p}_n(u)| \leq \frac{1}{2}C_{p''}\Delta^2 = \frac{1}{2}C_{p''}\frac{L^2}{n^2} \qquad \text{for all } u \in I.$$

**Sup-norm bound between $\tilde{p}_1$ and $\tilde{p}_n$.** Taking the supremum over $u \in I$ in the pointwise bounds above gives

$$\|\tilde{p}_1 - p\|_\infty \leq \tfrac{1}{2}C_{p''}L^2, \qquad \|p - \tilde{p}_n\|_\infty \leq \tfrac{1}{2}C_{p''}\frac{L^2}{n^2}.$$

By the triangle inequality in the sup norm,

$$\|\tilde{p}_1 - \tilde{p}_n\|_\infty \;\leq\; \|\tilde{p}_1 - p\|_\infty + \|p - \tilde{p}_n\|_\infty,$$

then,

$$\|\tilde{p}_1 - \tilde{p}_n\|_\infty \;\leq\; \|\tilde{p}_1 - p\|_\infty + \|p - \tilde{p}_n\|_\infty, \;\leq\; \tfrac{1}{2}C_{p''}L^2\Big(1 + \tfrac{1}{n^2}\Big).$$

$\square$

**Corollary A1** (Robustness of Theorem 2 under $n$-segment PWLA). *Suppose that, on the parameter range considered in this paper, the one-segment linear model $\tilde{p}_1$ used in Theorem 2 satisfies*

$$R(\tilde{p}_1)^2 \;\leq\; 1 - \delta_0$$

*for some margin $\delta_0 > 0$ (for example, under the condition $L\delta > \sigma^2$ in Theorem 2). Let $\tilde{p}_n$ be the $n$-segment piecewise-linear model defined above, and assume the conditions of Lemmas 1 and 2 hold. Then, for all $n \in \mathbb{N}$,*

$$\left|R(\tilde{p}_n)^2 - R(\tilde{p}_1)^2\right| \;\leq\; K\|\tilde{p}_n - \tilde{p}_1\|_\infty \;\leq\; \frac{1}{2}KC_{p''}L^2\Big(1 + \tfrac{1}{n^2}\Big),$$

*where $K > 0$ is the Lipschitz constant from Lemma 1 (measuring the sensitivity of $R(h)^2$ to perturbations of the weight $h$) and $C_{p''} := \sup_{u \in I}|p''(u)|$ is the curvature bound from Lemma 2 (quantifying how strongly the Gaussian weight can bend on the effective window). In particular, if the effective window satisfies*

$$KC_{p''}L^2 \;\leq\; \delta_0,$$

*then*

$$R(\tilde{p}_n)^2 \;\leq\; R(\tilde{p}_1)^2 + \frac{\delta_0}{2} \;\leq\; 1 - \frac{\delta_0}{2} \;<\; 1.$$

*Hence, the inequality $\mathrm{CV}_{\mathrm{asy}} < \mathrm{CV}_{\mathrm{sym}}$ derived under the one-segment linear model in Theorem 2 can be valid for any sufficiently accurate $n$-segment piecewise-linear approximation $\tilde{p}_n$ of the same Gaussian weight function on the effective window.*

Table A3: Results of accuracy and spike count under different deletion rates on CIFAR10 with VGG16.

| Methods | Ratio(%) | Acc.(%) | # of Spikes ($\times 10^3$) |
|---|---|---|---|
| *BOX* (Baseline) | 0 | 94.30 (-0.00) | 94 (-0.0%) |
| | 10 | 91.09 (-3.21) | 88 (-6.0%) |
| | 20 | 69.08 (-25.22) | 83 (-14.0%) |
| $A^2SG$ (Ours) | 0 | 95.29 (-0.00) | 89 (-0.0%) |
| | 10 | **92.59 (-2.81)** | **84 (-5.0%)** |
| | 20 | **69.08 (-24.89)** | **81 (-10.0%)** |

