# OpenReview forum: "A$^2$SG: Adaptive and Asymmetric Surrogate Gradients for Training Deep Spiking Neural Network"
_ICLR.cc/2026/Conference — Submitted to ICLR 2026_

### Official Review · Reviewer_wBJ2 · 2025-10-24

**Soundness:** 2
**Presentation:** 2
**Contribution:** 3
**Rating:** 6
**Confidence:** 4

**Summary:**

This paper introduces A$^2$SG, a unified framework designed to address the challenges of sharp loss landscapes and temporal inconsistency in deep SNN training, which arise from the use of surrogate gradients. The method employs a novel spatio-temporal adaptive strategy (ST-ASG) and a neuron-dynamics-aware asymmetric surrogate gradient (ASY) to guide the network towards flatter minima, thereby enhancing performance and generalization.

**Strengths:**

The work tackles a critical bottleneck in the SNN field. Its core contribution lies in establishing a theoretical link between the micro-level design of surrogate gradients (shape, window) and the macro-level geometry of the loss landscape (flatness). The effectiveness of the proposed method is substantiated by extensive and thorough experiments across a wide range of architectures and tasks, with convincing results.

**Weaknesses:**

1.A major flaw of this paper is the complete omission of the additional computational overhead introduced by A$^2$SG. The cost of running a Bayesian optimization search (Alg. A1) for every layer at each update interval appears to be substantial.

2.The theoretical foundation of the paper rests on strong assumptions that are not empirically validated. For instance, the analysis in Section 4.1 assumes a small local gradient variation $CV(\delta)$, and the proof of Thm. A2 relies on a linear approximation of a Gaussian distribution. The absence of discussion or experimental validation of these assumptions' validity during practical training weakens the persuasiveness of the theoretical contributions.

3.The authors apply strong data augmentation strategies (e.g., CutMix, RandAugment) for their method while comparing against SOTA results that appear to be cited directly from original papers. These baseline methods may not have been trained with the same level of augmentation, constituting an unfair comparison that obscures the true performance gain attributable to A$^2$SG itself.

4.The ablation study in Table 6 is conducted on CIFAR-10. Under strong data augmentation, the baseline performance is already very high, which diminishes the significance of A$^2$SG's marginal improvements. An ablation study on a more challenging dataset where the gains are larger (e.g., CIFAR-100) would be more convincing.

5.In Table 3, a special network architecture from E-SpikeFormer with T=1, D=4 is used. This is effectively an integer-activation network, not a conventional spiking network. The paper fails to explain how the surrogate gradient, which depends on membrane potential and a firing threshold, is applied and how β is searched in this non-binary, single-timestep setting. This leaves the reader confused about its application to this important model.

6.The adaptive process of A$^2$SG introduces several new, critical hyperparameters (e.g., search frequency $i_update$, number of observation points $n_obs$), yet the paper lacks a sensitivity analysis for them. This poses a significant challenge for reproducing the results.

**Questions:**

1.I strongly recommend that the authors provide a comparison of the wall-clock training time between A$^2$SG and a baseline method in the appendix to quantify the computational overhead.

2.To more robustly demonstrate the contribution of each component, I suggest adding an ablation study on the CIFAR-100 dataset

3.Please provide a detailed explanation of how A$^2$SG is adapted to the integer-activation model with T=1, D=4 in Table 3. What is the precise mechanism for applying the surrogate gradient and performing the β search in this scenario?

4.There are several undefined or unclear symbols in the paper, such as "CV" after Eq. 5 and the unit "in k" for spike counts in Table 6. Please proofread the entire manuscript carefully and correct these details.

---

> ### Author Response · Authors · 2025-12-02
>
> Thank you for your constructive feedback. We address your concerns and misunderstandings below.
>
> >W1. A major flaw of this paper is the complete omission of the additional computational overhead introduced by $A^2SG$. The cost of running a Bayesian optimization search (Alg. A1) for every layer at each update interval appears to be substantial.
>
> >Q1. I strongly recommend that the authors provide a comparison of the wall-clock training time between $A^2SG$ and a baseline method in the appendix to quantify the computational overhead.
>
> We report the overhead of our method in Table R1. Even if the search is performed at every epoch, the additional computation results in up to a 15% in training time. As the model size increases, the overhead tends to decrease. We have added this analysis to the revised manuscript, and it can be found in Table 6  (Section 5.4).
>
> **Table R1. Overhead analysis on CIFAR10.**
> | Architectures | Mehtods | Time per epoch (s) | $\Delta$ (%) |
> | --- | --- | --- | --- |
> | VGG16 | Baseline | 74 | +0 |
> |  | w/ S-ASG | 82 | +11 |
> |  | w/ T-ASG | 83 | +12 |
> |  | **$A^2SG$ (Ours)** | 85 | +15 |
> | ResNet19 | Baseline | 296  | +0 |
> |  | w/ S-ASG | 325 | +9 |
> |  | w/ T-ASG | 327 | +10 |
> |  | **$A^2SG$ (Ours)** | 338 | +14 |

---

> ### Author Response · Authors · 2025-12-02
>
> >W2. The theoretical foundation of the paper rests on strong assumptions that are not empirically validated. For instance, the analysis in Section 4.1 assumes a small local gradient variation CV($\delta$), and the proof of Thm. A2 relies on a linear approximation of a Gaussian distribution. The absence of discussion or experimental validation of these assumptions' validity during practical training weakens the persuasiveness of the theoretical contributions.
>
> Upon re-examining the assumption in Section 4.1 (”when the CV($\delta$) is small”), we found that the transition from Eq. (5) to Eq. (6) does not depend on the magnitude of CV($\delta$). The expression is obtained simply by substituting $\delta = \mu\mathbf{1} + \boldsymbol{\epsilon}$ and rearranging terms. Since this assumption is unnecessary and may cause confusion, we have removed it from the manuscript.
>
> The computation of the CV ratio in Thm. A2 is analytically intractable with the exact Gaussian distribution. The mean of the membrane-potential distribution is usually less than zero, due to the reset mechanism of spiking neurons. This fact ensures that $\mu < a = th-\beta$. Under this condition, because the effective window always exists as a narrow region to the right of the mean, the approximation to a monotonically decreasing linear function in this region can be valid.
>
> To address the reviewer’s concern more directly, we extend Thm. A2 from a single linear segment to a general n-segment piecewise linear approximation (PWLA).
>
> First, We additionally provide two lemmas as follows:
>
> >(Lemma 1) the squared CV ratio  $R(h)^2 := \left(\frac{\mathrm{CV}\{asy}(h)}{\mathrm{CV}\{sym}(h)} \right)^2$ changes at most proportionally to the sup-norm difference between $h$ and $\hat h$
>
> >(Lemma 2) the error between the one segment model $\tilde p_1$ used in Theorem 2 and an n-segment PWLA approximation $\tilde p_n$ can be bounded explicitly via Taylor’s theorem
>
> Combining these, we obtain a corollary. If the one-segment model satisfies $R(\tilde p_1)^2 \le 1 - \delta_0$ for some margin $(\delta_0>0\)$, and the effective window is not too wide (practically less than 1), then the same strict inequality $R(\tilde p_n)^2 < 1$ continues to hold for any sufficiently accurate n-segment PWLA approximation $\tilde p_n$ of the Gaussian on the effective window. We experimentally verified that this ratio is less than 1. We added the details in the appendix of the revised manuscript (section A.15).
>
> Furthermore, we provide additional experiments evidence supporting our assumption. Specifically, we report the values of $L\kappa$ and $\sigma^2$ across training epochs, as shown in Table R2. These results correspond to the first and last convolution layers in VGG16 trained on CIFAR10. Both $L\kappa$ and $\sigma^2$ were computed by averaging their values across all time steps. According to Table R2, the condition $L\kappa$ > $\sigma^2$ stated in Theorem 2 is not satisfied during the early stages of training; however, as training progresses, the condition becomes satisfied and holds for nearly all layers and time steps. These observations demonstrate that the linear approximation of the Gaussian assumption remains valid in practice. We have added this discussion to the revised manuscript, which can be found at line 306 and in Figure 4 (Section 4.3).
>
> **Table R2. Value of $L\kappa$ and $\sigma^2$ across training epochs.**
> | Epoch | Conv1 $L\kappa$  | Conv1 $\sigma^2$ | Conv5-2 $L\kappa$ | Conv5-2 $\sigma^2$ |
> | --- | --- | --- | --- | --- |
> | 11 | 0.939 | 1.220 | 0.998 | 1.283 |
> | 100 | 0.534 | 0.441 | 0.424 | 0.121 |
> | 150 | 0.517 | 0.473 | 0.419 | 0.120 |
> | 200 | 0.489 | 0.462 | 0.425 | 0.108 |
> | 250 | 0.483 | 0.430 | 0.430 | 0.098 |
> | 310 | 0.474 | 0.406 | 0.432 | 0.096 |
>
> >W3. The authors apply strong data augmentation strategies (e.g., CutMix, RandAugment) for their method while comparing against SOTA results that appear to be cited directly from original papers. These baseline methods may not have been trained with the same level of augmentation, constituting an unfair comparison that obscures the true performance gain attributable to $A^2SG$ itself.
>
> Regarding the augmentation, we note that the most recent works cited in Table 2 of our manuscript use the same augmentation method as ours, namely AutoAugmentation (IM (Guo et al., 2022), RMP (Guo et al., 2023a), ShortcutBP (Guo et al., 2024), MPD-AGL (Jiang et al., 2025)). Therefore, using similar augmentation strategies in our experiments ensures a fair comparison with these methods. When we compare the two most recent SOTA-level studies (ShortcutBP (Guo et al., 2024), MPD-AGL (Jiang et al., 2025)), we can confirm the effectiveness of our approach.

---

> ### Author Response · Authors · 2025-12-02
>
> >W4. The ablation study in Table 6 is conducted on CIFAR10. Under strong data augmentation, the baseline performance is already very high, which diminishes the significance of $A^2SG$'s marginal improvements. An ablation study on a more challenging dataset where the gains are larger (e.g., CIFAR100) would be more convincing.
>
> >Q2. To more robustly demonstrate the contribution of each component, I suggest adding an ablation study on the CIFAR100 dataset.
>
> As the reviewer pointed out, on CIFAR10, the strong data augmentation already leads to a high baseline performance, so the performance improvements may appear marginal. Nevertheless, our method is still meaningful in that it provides consistent performance gains across various datasets and models. In addition, following your suggestion, we conducted further ablation experiments on the more challenging CIFAR100 dataset. Table R3 clearly shows the contributions of the adaptive and asymmetric components. Applying the spatial adaptive surrogate gradient (S-ASG) improves accuracy while reducing the spike count, whereas applying only the temporal adaptive surrogate gradient (T-ASG) enhances accuracy with a slight increase in spike count. Their combination, ST-ASG and the $A^2SG$, with an asymmetric surrogate additionally incorporated, achieves the highest accuracy of 95.29\%, while maintaining a similar spike count. It thereby confirms the effectiveness of integrating all components. Moreover, these results exhibit a trend consistent with CIFAR10 while showing larger performance gains on the more difficult dataset (CIFAR10: +0.47\%, CIFAR100: +1.31\%) as the reviewer expected. This provides additional evidence for the effectiveness of our method. We have added these experimental results to Section 5.4 ‘Ablation Studies’ of the revised manuscript.
>
> **Table R3. Ablation study on CIFAR100 with VGG16, comparing spatial (S-ASG), temporal (T-ASG), spatio-temporal adaptation (ST-ASG), and ST-ASG with $ASY$ ($A^2SG$).**
> | Methods            | Acc. (%)      | \# of Spikes ($\times10^3$) | Latency (sec/epoch) |
> |--------------------|------------------|--------------------|---------------------|
> | $BOX$ (Baseline)   | 74.24$\pm$0.08      | 100.3$\pm$1.2         | 74 (+0%)            |
> | w/ S-ASG           | 74.57$\pm$0.08       | 93.2$\pm$0.7         | 82 (+11%)           |
> | w/ T-ASG           | 74.54$\pm$0.06       | 104.9$\pm$2.2         | 83 (+12%)           |
> | w/ ST-ASG          | 74.73$\pm$0.07       | 100.2$\pm$0.2         | 85 (+15%)           |
> | **$A^2SG$ (Ours)**    | **75.21$\pm$0.08**   | **100.2$\pm$0.4**       | **85 (+15%)**       |

---

> ### Author Response · Authors · 2025-12-02
>
> >W5. In Table 3, a special network architecture from E-SpikeFormer with $T=1$, $D=4$ is used. This is effectively an integer-activation network, not a conventional spiking network. The paper fails to explain how the surrogate gradient, which depends on membrane potential and a firing threshold, is applied and how $\beta$ is searched in this non-binary, single-timestep setting. This leaves the reader confused about its application to this important model.
>
> >Q3. Please provide a detailed explanation of how $A^2SG$ is adapted to the integer-activation model with $T=1$, $D=4$ in Table 3. What is the precise mechanism for applying the surrogate gradient and performing the $\beta$ search in this scenario?
>
> As the reviewer noted, TGC is generally applicable when the time step is greater than one. However, because spatio-temporal dynamics are an essential property of SNNs, the effective time step in SNNs is typically greater than one. Thus, our approach can be applied to most SNN models.
>
> For I-LIF, as used in E-SpikeFormer, the effective time step is $T$$\times$$D$ rather than $T$. E-SpikeFormer adopted integer LIF neurons with multiple thresholds, which is distinct from the conventional SNNs with LIF neurons. To efficiently train large-scale models on GPUs, it uses integer values as a substitute for the temporal spike trains. Thus, this mechanism implicitly incorporates temporal dynamics even if $T=1$. The model can be regarded as operating with an implicit time step equal to $T$$\times$$D$, rather than a single time step.
>
> Based on this perspective, we aligned the time step of the conventional LIF neurons with the integer activation value of I-LIF. For example, when $T$$\times$$D$ is $1$$\times$$4$ for I-LIF, we applied S-ASG to neurons with an activation value of four, corresponding to the last time step. We then sequentially applied T-ASG to neurons with activation values of three, two, and one.
>
> For the effective window ($\beta$), we set it to be centered on each threshold, as in LIF with a single threshold. For example, if $\textrm{th}_i$ is a threshold at integer $i$, the effective window of $\textrm{th}_i$ is set to $[\textrm{th}_i-\beta_i, \textrm{th}_i+\beta_i]$. From this state, we changed $\beta_i$ through our adaptive method. This is how we implemented our approach in I-LIF neurons. We have added this explanation to the revised manuscript at line 421 (Section 5.3).
>
> Additionally, E-SpikeFormer relies on STE to pass gradients, which can be regarded as a form of surrogate gradient. However, STE is designed primarily for quantization-aware training and does not consider SNN-specific dynamics. In contrast, our approach provides an SNN-aware surrogate gradient. In this regard, our method is also beneficial for I-LIF neurons, which have been widely adopted in SNN-ViT architectures.

---

> ### Author Response · Authors · 2025-12-02
>
> >W6. The adaptive process of $A^2SG$ introduces several new, critical hyperparameters (e.g., search frequency $i_{update}$, number of observation points $n_{opb}$), yet the paper lacks a sensitivity analysis for them. This poses a significant challenge for reproducing the results.
>
> We present additional analysis on the sensitivity of update frequency and the hyperparameters used in Bayesian optimization, specifically ($n_{obs}$, $n_{eval}$, and $\delta$). The results reported in the manuscript are based on a default configuration where the update frequency is set to 1 epoch, ($n_{obs}$, $n_{eval}$) is set to (100, 150), and $\delta$ is set to 0.05.
>
> First, to assess the sensitivity of the update frequency, we conducted experiments by increasing it to 50 or 100 epochs. As shown in Table R4, increasing the update frequency resulted in accuracy drops of about 0.2%.
>
> Next, we investigated the sensitivity of the Bayesian optimization hyperparameters by varying ($n_{obs}$, $n_{eval}$) to (10, 15) and (300, 450). The results show that increasing ($n_{obs}$, $n_{eval}$) led to slight increases in both accuracy.
>
> Finally, we examined the sensitivity of $\delta$, which represents the search width around the current $\beta$. The results for $\delta = 0.01$ and $\delta = 0.1$ are reported in Table R4.
>
> Overall, these experiments indicate that $A^2SG$ is not overly sensitive to either the update frequency or the Bayesian optimization hyperparameters.
>
> We have included a new section (5.7 Sensitivity Analysis) in the revised manuscript to present this analysis.
>
> These sensitivity analysis results affirm that our method is not significantly affected by specific parameters.
>
> **Table R4. Comparison with different $\beta$ update frequencies (epoch) and Bayesian optimization hyperparameter ($n_{obs}$, $n_{eval}$, $\delta$) settings on CIFAR10 with VGG16.**
> | Update frequency  | $n_{obs}$ | $n_{eval}$ | $\delta$ | Accuracy (%) |
> | --- | --- | --- | --- | --- |
> | 1 epoch | 100 | 150 | 0.05 | 95.29$\pm$0.04 |
> | 50 epoch | 100 | 150 | 0.05 | 95.10$\pm$0.02 |
> | 100 epoch | 100 | 150 | 0.05 | 95.06$\pm$0.06 |
> |  |  |  |  |  |  |
> | 1 epoch | 10 | 15 | 0.05 | 95.10$\pm$0.06 |
> | 1 epoch | 100 | 150 | 0.05 | 95.29$\pm$0.04 |
> | 1 epoch | 300 | 450 | 0.05 | 95.31$\pm$0.02 |
> |  |  |  |  |  |  |
> | 1 epoch | 100 | 150 | 0.01 | 95.15$\pm$0.08 |
> | 1 epoch | 100 | 150 | 0.05 | 95.29$\pm$0.04 |
> | 1 epoch | 100 | 150 | 0.10 | 95.22$\pm$0.03 |
>
> >Q4. There are several undefined or unclear symbols in the paper, such as "CV" after Eq. 5 and the unit "in k" for spike counts in Table 6. Please proofread the entire manuscript carefully and correct these details.
>
> Thank you for pointing this out. We have corrected it in the manuscript and uploaded the revised version accordingly.
>
> We thank the reviewer once again for the valuable comments, and we hope that our responses adequately address the concerns and misunderstandings.

---

### Official Review · Reviewer_YJia · 2025-10-25

**Soundness:** 2
**Presentation:** 2
**Contribution:** 2
**Rating:** 4
**Confidence:** 4

**Summary:**

This work proposes an adaptive and asymmetric calculation scheme for surrogate gradients of SNNs, which is based on the theoretical analysis of local gradient variation and the curvature of the loss landscape.

**Strengths:**

1. The authors discuss their adaptive control strategies for the surrogate gradient interval from the perspectives of spatial gradient variation (SGV) and temporal gradient consistency (TGC). The overall argumentation process is logical.

**Weaknesses:**

1. Since this work considers issues such as temporal gradient consistency (TGC) when calculating surrogate gradients, performance validation on time-series datasets is crucial. However, in Tab. 4, the performance improvement of $A^2SG$ compared to STBP-tdBN (baseline) on CIFAR10-DVS is not significant (81.68% v.s. 81.30%, < 1%).

2. For large-scale datasets (e.g. ImageNet-1k), the performance improvement achieved by this work in Tab. 3 is also less than 1%. In addition, using E-SpikeFormer as the baseline model may not be suitable because it considers a multi-threshold spike firing mechanism and combines other optimization techniques, which cannot fully reflect the impact of surrogate gradient calculation on performance. On the contrary, in some relatively simple network structures, the influence of surrogate gradients may be more prominent.

3. In recent years, researchers have attempted to further enhance the performance of SNNs from various perspectives, such as proposing advanced spiking models, designing spiking self-attention calculation schemes, introducing loss functions and BN modules based on temporal information, etc. The improvement of surrogate gradient calculation is merely one of these optimization routes. Therefore, I tend to think that the overall contribution of this work to the SNN community is relatively limited.

**Questions:**

See Weaknesses Section.

---

> ### Author Response · Authors · 2025-11-24
>
> Thank you for your constructive feedback. We address your concerns and misunderstandings below.
>
> >W1. Since this work considers issues such as temporal gradient consistency (TGC) when calculating surrogate gradients, performance validation on time-series datasets is crucial. However, in Tab. 4, the performance improvement of $A^2SG$ compared to STBP-tdBN (baseline) on CIFAR10-DVS is not significant (81.68% v.s. 81.30%, < 1%).
>
> Before addressing the reviewer's concerns, we would like to clarify an oversight in our manuscript. During the development of the method and experiments, we parameterized the constant 0.5 in Equation (10) and used different values depending on the model or datasets. However, this modification was not reflected in the submitted manuscript. We sincerely apologize for the confusion. The equation in the manuscript has been corrected accordingly.
>
> In our initial submission, we did not explore the value of $h$ for the neuromorphic dataset. Instead, we used the same settings for VGG16 and CIFAR10.
> Even without optimization for the DVS dataset, our method performs comparably to state-of-the-art SOTA results (R1, R2) using the same model and similar settings, such as time steps (4 or 5).
> Despite the potential for differences in gradient sparsity and magnitude due to differences in the characteristics of frame-based and neuromorphic datasets, our method was able to achieve good performance on the neuromorphic dataset even with the frame-based dataset settings.
>
> During this discussion phase, we aimed to optimize the hyperparameter for CIFAR10-DVS based on the reviewers' comments. We used grid search to vary the value of $h$ from 0.5 to 0.8 in increments of 0.1.
> Additionally, we tested two additional settings of $\pm$0.05 near the value that produced the highest accuracy.
>
> As a result, we improved the accuracy to 82.36%, and absolute gain of more than 1% compared to our baseline (tdBN). The results for different values of $h$ are summarized in Table R1. We also note that CIFAR10-DVS is widely regarded as one of the most challenging neuromorphic datasets. The reported accuracy, achieved with VGGSNN under the four time steps settings, is to the best of our knowledge at the state-of-the-art level, compared to the recent works [R1, R2]. This demonstrates that our method remains effective also in neuromorphic datasets. The updated results are presented in Table 4 of the manuscript.
> For future work, we plan to propose a method for parameter setting that accounts for gradient sparsity and size.
>
> **Table R1. Accuracy comparison on CIFAR10-DVS with respect to $h$.**
> | Dataset | Architecture | Methods|Time steps | $h$ | Acc. (%) |
> |---|---|------|----------------| --- | --- |
> |CIFAR10-DVS|VGGSNN|  HSD [R1]| 5|  - | 81.10 |
> |||TMC [R2]| 4|-  | 81.76   |
> |||$A^2SG$| 4|0.60  | 81.68$\pm$0.37   |
> ||| |4|0.70  | 81.72$\pm$0.02   |
> ||| |4|0.75 | 82.36$\pm$0.01   |
> ||| |4|0.80  | 80.29$\pm$0.03   |

---

> ### Author Response · Authors · 2025-11-24
>
> >W2. For large-scale datasets (e.g. ImageNet-1k), the performance improvement achieved by this work in Tab. 3 is also less than 1%. In addition, using E-SpikeFormer as the baseline model may not be suitable because it considers a multi-threshold spike firing mechanism and combines other optimization techniques, which cannot fully reflect the impact of surrogate gradient calculation on performance. On the contrary, in some relatively simple network structures, the influence of surrogate gradients may be more prominent.
>
> Regarding the concern about the suitability of the E-SpikeFormer baseline, we would like to clarify that although E-SpikeFormer incorporates a multi-threshold spike firing mechanism, this does not conflict with surrogate-gradient-based training, which means our method is applicable. As such, our method is applicable to any neuron type and models architectures as long as gradient-based training is used. To validate this fact and verify effectiveness on SOTA-level model, we chose E-SpikeFormer for ImageNet datasets.
>
> To apply our method to E-SpikeFormer (I-LIF neuron), we aligned the time step of the conventional LIF neurons with the integer activation value of I-LIF. For example, when $T$$\times$$D$ is $1$$\times$$4$ for I-LIF, we applied S-ASG to neurons with an activation value of four, corresponding to the last time step. We then sequentially applied T-ASG to neurons with activation values of three, two, and one.
>
> For the effective window ($\beta$), we set it to be centered on each threshold, as in LIF with a single threshold. For example, if $\textrm{th}_i$ is a threshold at integer $i$, the effective window of $\textrm{th}_i$ is set to $[\textrm{th}_i-\beta_i, \textrm{th}_i+\beta_i]$. From this state, we changed $\beta_i$ through our adaptive method. This is how we implemented our approach in I-LIF neurons. We have added this explanation to the revised manuscript at lines 421 (Section 5.3).
>
> Nevertheless, as the reviewer suggested, we are preparing additional experiments on simpler architectures, such as ResNet. However, ImageNet experiments require substantially more computation time than our other experiments, and because many additional experiments requested during the rebuttal period are already consuming our available resources, it is uncertain whether we will be able to provide the ImageNet results within the discussion period. However, $A^2SG$ has shown consistent improvements in our ResNet-based experiments on CIFAR10 and CIFAR100, and we are confident that the simpler architecture (ResNet) will also improve performance on ImageNet.
>
> We will make our best effort to provide the results as soon as possible. Even if some experiments cannot be completed during the discussion period, we will continue experimenting with them and incorporate the results into the revised manuscript.

---

> ### Author Response · Authors · 2025-12-02
>
> >W3. In recent years, researchers have attempted to further enhance the performance of SNNs from various perspectives, such as proposing advanced spiking models, designing spiking self-attention calculation schemes, introducing loss functions and BN modules based on temporal information, etc. The improvement of surrogate gradient calculation is merely one of these optimization routes. Therefore, I tend to think that the overall contribution of this work to the SNN community is relatively limited.
>
> As reviewer ‘WK4e’ mentioned, we also believe that research on surrogate functions remains one of the core challenges in SNNs. While surrogate gradients represent only one of the many efforts to improve SNN performance, our method is broadly applicable to any spiking neuron model, architecture, or loss function that relies on gradient-based training. We further demonstrate that our approach can even be applied to integer spiking neurons trained with STE (E-SpikeFormer). Since most gradient-based SNN training methods inherently depend on surrogate gradients due to the non-differentiability of the spiking neurons, our method can serve as a generally applicable enhancement. In addition, we conducted further experiments to demonstrate that our method can be applied to other loss functions. Table R2 shows consistent improvements in both performance and efficiency with RMP-loss(Guo et al. ,2023a), similar to the tdBN baseline, and this experiment is reflected in section 5.6 of the revised manuscript. Based on our empirical results, we strongly believe that our approach can improve the training performances across various lines of research with gradient-based training in deep SNNs.
>
> **Table R2. Comparison between tdBN and RMP-loss on CIFAR100 with VGG16.**
> | Dataset | Methods | Acc. (%) | \# of Spikes ($\times10^3$) |
> | --- | --- | --- | --- |
> | CIFAR100 | tdBN | 74.24$\pm$0.08 | 100.3$\pm$1.2 |
> |  | RMP | 74.39$\pm$0.07 | 102.8$\pm$2.9 |
> |  | tdBN + $A^2SG$ | 75.21$\pm$0.08 | 100.2$\pm$0.4 |
> |  | RMP + $A^2SG$| 75.25$\pm$0.03 | 102.3$\pm$1.1 |
>
> We thank the reviewer once again for the valuable comments, and we hope that our responses adequately address the concerns and misunderstandings.
>
> [R1] Zhong, Xian, et al. "Towards low-latency event-based visual recognition with hybrid step-wise distillation spiking neural networks." *Proceedings of the 32nd ACM international conference on multimedia*. 2024.
>
> [R2] Jiaqi Yan, Changping Wang, De Ma, Huajin Tang, Qian Zheng, and Gang Pan. Training high performance spiking neural network by temporal model  calibration. In Forty-second International Conference on Machine Learning, 2025.

---

### Official Review · Reviewer_uKH8 · 2025-10-28

**Soundness:** 3
**Presentation:** 2
**Contribution:** 2
**Rating:** 4
**Confidence:** 3

**Summary:**

This paper propose a method of surrogate gradients to improve SNN backpropagation training. This method dynamically adjusts the surrogate gradient window width for spatio-temporal adaptation and introduces an asymmetric surrogate that assigns larger gradients to neurons with higher membrane potentials. The authors provide theoretical analysis linking gradient variation to loss landscape curvature and demonstrate consistent improvements across diverse architectures and tasks.

**Strengths:**

- Theoretical analysis between relationship of surrogate gradient design to loss landscape sharpness, asymmetric surrogates to CV
- New design of surrogate gradient where larger gradients were assigned to neurons with higher membrane potentials
- Comprehensive experimental validation on image classification and segmentation datasets and models

**Weaknesses:**

- Bayesian optimization for β search runs every epoch. This computational overhead was not analyzed. No wall-clock training time comparisons provided.
- Performance improvement is limited. This may imply that this question is not a hard core problem for snn training.
- No study on $\beta$ update frequency (every epoch vs every N epochs). Sensitivity to Bayesian optimization hyperparameters (nobs, neval, $\delta$)?
- How about other neuron models (PLIF, ALIF, etc.)? Please verify this model apply to them too.

**Questions:**

- Line 884: "β fixed to 0.5" but then say "optimal β searched every epoch"?

---

> ### Author Response · Authors · 2025-11-24
>
> Thank you for your constructive feedback. We address your concerns and misunderstandings below.
>
> >W1. Bayesian optimization for $\beta$ search runs every epoch. This computational overhead was not analyzed. No wall-clock training time comparisons provided.
>
> We report the overhead of our method in Table R1. Even if the search is performed at every epoch, the additional computation results in up to a 15% in training time. As the model size increases, the overhead tends to decrease. We have added this analysis to the revised manuscript, and it can be found in Table 6  (Section 5.4).
>
> **Table R1. Overhead analysis on CIFAR-10.**
> | Architectures | Mehtods | Time per epoch (s) | $\Delta$ (%) |
> | --- | --- | --- | --- |
> | VGG16 | Baseline | 74 | +0 |
> |  | w/ S-ASG | 82 | +11 |
> |  | w/ T-ASG | 83 | +12 |
> |  | $A^2SG$ **(Ours)** | 85 | +15 |
> | ResNet19 | Baseline | 296  | +0 |
> |  | w/ S-ASG | 325 | +9 |
> |  | w/ T-ASG | 327 | +10 |
> |  | $A^2SG$ **(Ours)** | 338 | +14 |
>
> >W2. Performance improvement is limited. This may imply that this question is not a hard core problem for snn training.
>
> Although the performance gains may appear marginal, this is largely because our experiments are conducted on models that already achieve near state-of-the-art performance. Importantly, our method does not target specific tasks to improve benchmark performance. It aims to adjust gradients to improve performance regardless of the model or task. In this regard, as reviewer ‘WK4e’ noted, research on surrogate gradient remains one of the core challenges in SNNs. Our method is applicable to any model architecture trained with gradient-based algorithms, such as STBP, and we further demonstrate that it can even be applied to various types of neurons, including I-LIF and P-LIF.  In addition, to verify the compatibility of our approach with other learning methods, we applied $A^2SG$ to RMP-loss (Guo et al. ,2023a). Table R2 shows consistent improvements in both performance and efficiency, similar to the tdBN baseline. This experiment is reflected in section 5.6 of the revised manuscript. These results suggest that our approach can complement existing efforts to improve SNN performance and has the potential to benefit the broader SNN community.
>
> **Table R2. Comparison between tdBN and RMP-loss on CIFAR100 with VGG16.**
> | Dataset | Methods | Acc. (%) | \# of Spikes ($\times10^3$) |
> | --- | --- | --- | --- |
> | CIFAR100 | tdBN | 74.24$\pm$0.08 | 100.3$\pm$1.2 |
> |  | RMP | 74.39$\pm$0.07 | 102.8$\pm$2.9 |
> |  | tdBN + $A^2SG$ | 75.21$\pm$0.08 | 100.2$\pm$0.4 |
> |  | RMP + $A^2SG$| 75.25$\pm$0.03 | 102.3$\pm$1.1 |
>
> >W3. No study on $\beta$ update frequency (every epoch vs every N epochs). Sensitivity to Bayesian optimization hyperparameters (nobs, neval, $\delta$)?
>
> We present additional analysis on the sensitivity of update frequency and the hyperparameters used in Bayesian optimization, specifically ($n_{obs}$, $n_{eval}$, and $\delta$). The results reported in the manuscript are based on a default configuration where the update frequency is set to 1 epoch, ($n_{obs}$, $n_{eval}$) is set to (100, 150), and $\delta$ is set to 0.05.
>
> First, to assess the sensitivity of the update frequency, we conducted experiments by increasing it to 50 or 100 epochs. As shown in Table R3, increasing the update frequency resulted in accuracy drops of about 0.2%.
>
> Next, we investigated the sensitivity of the Bayesian optimization hyperparameters by varying ($n_{obs}$, $n_{eval}$) to (10, 15) and (300, 450). The results show that increasing ($n_{obs}$, $n_{eval}$) led to slight increases in both accuracy.
>
> Finally, we examined the sensitivity of $\delta$, which represents the search width around the current $\beta$. The results for $\delta = 0.01$ and $\delta = 0.1$ are reported in Table R3.
>
> Overall, these experiments indicate that $A^2SG$ is not overly sensitive to either the update frequency or the Bayesian optimization hyperparameters.
>
> We have included a new section (5.7 Sensitivity Analysis) in the revised manuscript to present this analysis.
>
> These sensitivity analysis results affirm that our method is not significantly affected by specific parameters.
>
> **Table R3. Comparison with different $\beta$ update frequencies (epoch) and Bayesian optimization hyperparameter ($n_{obs}$, $n_{eval}$, $\delta$) settings on CIFAR10 with VGG16.**
> | Update frequency  | $n_{obs}$ | $n_{eval}$ | $\delta$ | Acc. (%) |
> | --- | --- | --- | --- | --- |
> | 1 epoch | 100 | 150 | 0.05 | 95.29$\pm$0.04 |
> | 50 epoch | 100 | 150 | 0.05 | 95.10$\pm$0.02 |
> | 100 epoch | 100 | 150 | 0.05 | 95.06$\pm$0.06 |
> |  |  |  |  |  |  |
> | 1 epoch | 10 | 15 | 0.05 | 95.10$\pm$0.06 |
> | 1 epoch | 100 | 150 | 0.05 | 95.29$\pm$0.04 |
> | 1 epoch | 300 | 450 | 0.05 | 95.31$\pm$0.02 |
> |  |  |  |  |  |  |
> | 1 epoch | 100 | 150 | 0.01 | 95.15$\pm$0.08 |
> | 1 epoch | 100 | 150 | 0.05 | 95.29$\pm$0.04 |
> | 1 epoch | 100 | 150 | 0.10 | 95.22$\pm$0.03 |

---

> ### Author Response · Authors · 2025-11-24
>
> >W4. How about other neuron models (PLIF, ALIF, etc.)? Please verify this model apply to them too.
>
> To demonstrate that our method is applicable beyond the LIF neuron, we experimented on I-LIF (E-SpikeFormer). To further validate the effectiveness, we additionally implemented PLIF [R1] and applied our approach. The results are presented in the Table R4. The LIF results are identical to those used in Table 6 of the manuscript, and for fairness, the PLIF experiments were also repeated four times. These results show that our method is not limited to LIF but can be applied to a variety of neuron models. We have added these experimental results to the revised manuscript in Section 5.6.
>
> **Table R4. Performance comparison between LIF and PLIF.**
> | Dataset | Architecture | Methods | Neuron models | Acc. (%) | \# of Spikes ($\times10^3$)|
> | --- | --- | --- | --- | --- | --- |
> | CIFAR10 | VGG16 | Baseline | LIF | 94.84$\pm$0.05 | 94.6$\pm$1.0|
> |  |  | **$A^2SG$ (Ours)** | LIF | 95.29$\pm$0.04 | 84.9$\pm$1.8|
> |  |  | Baseline | PLIF | 94.99$\pm$0.03 | 91.6$\pm$7.4|
> |  |  | **$A^2SG$ (Ours)** | PLIF  | 95.33$\pm$0.05 | 82.2$\pm$0.5|
>
> >Q1.  Line 884: "β fixed to 0.5" but then say "optimal β searched every epoch"?
>
> We apologize for the confusion. The intended meaning was that $\beta$ is initialized to 0.5, not fixed at 0.5. After initialization, $\beta$ is updated at each epoch through the search strategy. Thank you for pointing this out. We have revised the corresponding part of the manuscript accordingly, and the corrected sentence can be found in Section A.10. (The correct statement: “For all experiments, the effective window $\beta$ was initialized to 0.5, and the optimal $\beta$ during training was searched at the first iteration of every epoch.”)
>
> We thank the reviewer once again for the valuable comments, and we hope that our responses adequately address the concerns and misunderstandings.
>
> [R1] Fang, Wei, et al. "Incorporating learnable membrane time constant to enhance learning of spiking neural networks." *Proceedings of the IEEE/CVF international conference on computer vision*. 2021.

---

### Official Review · Reviewer_WK4e · 2025-10-30

**Soundness:** 2
**Presentation:** 3
**Contribution:** 3
**Rating:** 4
**Confidence:** 4

**Summary:**

The paper addresses sharp loss landscapes and inconsistent gradients across time steps in training spiking neural networks by proposing adaptive and asymmetric surrogate gradients. At the final time step, the method minimizes spatial gradient variability (SGV) to stabilize updates. At earlier time steps, it maximizes temporal gradient consistency (TGC) to align update directions. The asymmetric surrogate assigns larger gradients to membrane potentials closer to the firing threshold. Together, these choices maintain low SGV and high TGC throughout training and are associated with a smaller maximum eigenvalue of the Fisher information matrix, steering optimization toward flatter minima and better generalization. Theoretical analysis supports the design, and experiments across multiple architectures and datasets show consistent improvements over baselines.

**Strengths:**

1.	The method targets core challenges in SNN training and is supported by clear, rigorous analysis and proofs, yielding coherent and well-substantiated conclusions.

2.	Extensive experiments and ablations across diverse datasets and architectures consistently confirm the method’s effectiveness over baselines.

**Weaknesses:**

1.	Theorem 1 assumes Gaussian membrane potentials with mean below $a$. In practice, layer and time-dependent distributions may violate this and drift during training.

2.	The paper does not report the overhead introduced by Bayesian search for $\beta$


3.	The gains over STBP on neuromorphic datasets are not obvious. This may suggest only moderate performance on neuromorphic data. More evaluations across additional models and neuromorphic benchmarks are needed to substantiate the method’s effectiveness in this setting.

4.	TGC is only meaningful when T>1. If configurations such as E-SpikeFormer with $A^2SG$ use T=1, it is difficult to establish the effectiveness of the proposed method.

**Questions:**

1.	Why SGV be measured at the final time step rather than at other time steps?

2.	How large is the practical gain from Bayesian search? How much improvement does it deliver over using a fixed $\beta$, a simple grid search, or other methods?

I would raise my score if the authors can address these weaknesses and questions.

---

> ### Author Response · Authors · 2025-11-24
>
> Thank you for your constructive feedback. We address your concerns and misunderstandings below.
>
> >W1. Theorem 1 assumes Gaussian membrane potentials with mean below $a$. In practice, layer and time-dependent distributions may violate this and drift during training.
>
> First, contrary to the reviewer's concern, the average membrane potential remains significantly smaller than the conventional parameter $a$ (0.5). Because of the reset property of spiking neurons, the mean of the membrane potential distribution is usually less than zero in most cases. Even if it increases, it reached up to about 0.23 in our experiments, which is much less than the parameter $a$. Therefore, even if the membrane potential drifts during training, as the reviewer’s concern, our assumptions are rarely violated.
>
> For more concrete evidence, we additionally provide the value of $L\kappa$ and $\sigma^2$ across training epochs, as shown in Table R1.  These results correspond to the first and last convolution layers in VGG16 trained on CIFAR10. $L\kappa$ and $\sigma^2$ were computed by averaging their values across all time steps. As shown in Table R1, the condition $L\kappa$ > $\sigma^2$ stated in **Theorem 2** is not satisfied during the very early phase of training; however, as training proceeds, the condition becomes satisfied and generally holds across layers and time steps. In conclusion, the assumption in the theorem holds in our experiments, and contrary to your concern, the model does not exhibit drift; instead, the condition becomes increasingly well-aligned as training progresses. We have added a discussion of these observations to the revised manuscript, and the updated content can be found at line 305 and Figure 4 (in Section 4.3).
>
> **Table R1. Value of $L\kappa$ and $\sigma^2$ across training epochs.**
>
> | Epoch | Conv1 $L\kappa$  | Conv1 $\sigma^2$ | Conv5-2 $L\kappa$ | Conv5-2 $\sigma^2$ |
> | --- | --- | --- | --- | --- |
> | 11 | 0.939 | 1.220 | 0.998 | 1.283 |
> | 100 | 0.534 | 0.441 | 0.424 | 0.121 |
> | 150 | 0.517 | 0.473 | 0.419 | 0.120 |
> | 200 | 0.489 | 0.462 | 0.425 | 0.108 |
> | 250 | 0.483 | 0.430 | 0.430 | 0.098 |
> | 310 | 0.474 | 0.406 | 0.432 | 0.096 |
>
> >W2. The paper does not report the overhead introduced by Bayesian search for $\beta$
>
> We report the overhead of our method in Table R2. Even if the search is performed at every epoch, the additional computation results in up to a 15% in training time. As the model size increases, the overhead tends to decrease. We have added this analysis to the revised manuscript, and it can be found in Table 6  (Section 5.4).
>
> **Table R2. Overhead analysis on CIFAR10.**
> |Architectures|Mehtods|Time per epoch (s) | $\Delta$ (%) |
> | --- | --- | --- | --- |
> | VGG16 | Baseline | 74 | +0 |
> || w/ S-ASG | 82 | +11 |
> || w/ T-ASG | 83 | +12 |
> || $A^2SG$ **(Ours)** | 85 | +15 |
> |ResNet19 | Baseline |296| +0 |
> || w/ S-ASG|325|+9|
> || w/ T-ASG|327|+10|
> || $A^2SG$ **(Ours)**|338|+14|

---

> ### Author Response · Authors · 2025-11-24
>
> >W3. The gains over STBP on neuromorphic datasets are not obvious. This may suggest only moderate performance on neuromorphic data. More evaluations across additional models and neuromorphic benchmarks are needed to substantiate the method’s effectiveness in this setting.
>
> Before addressing the reviewer's concerns, we would like to clarify an oversight in our manuscript. During the development of the method and experiments, we parameterized the constant 0.5 in Equation (10) and used different values depending on the model or datasets. However, this modification was not reflected in the submitted manuscript. We sincerely apologize for the confusion. The equation in the manuscript has been corrected accordingly.
>
> In our initial submission, we did not explore the value of $h$ for the neuromorphic dataset. Instead, we used the same settings for VGG16 and CIFAR10.
> Even without optimization for the DVS dataset, our method performs comparably to state-of-the-art (SOTA) results (R1, R2) using the same model and similar settings, such as time steps (4 or 5).
> Despite the potential for differences in gradient sparsity and magnitude due to differences in the characteristics of frame-based and neuromorphic datasets, our method was able to achieve good performance on the neuromorphic dataset even with the frame-based dataset settings.
>
> During this discussion phase, we aimed to optimize the hyperparameter for CIFAR10-DVS based on the reviewers' comments. We used grid search to vary the value of $h$ from 0.5 to 0.8 in increments of 0.1.
> Additionally, we tested two additional settings of $\pm$0.05 near the value that produced the highest accuracy.
>
> As a result, we improved the accuracy to 82.36%, and absolute gain of more than 1% compared to our baseline (tdBN). The results for different values of $h$ are summarized in Table R3. We also note that CIFAR10-DVS is widely regarded as one of the most challenging neuromorphic datasets. The reported accuracy, achieved with VGGSNN under the four time steps settings, is to the best of our knowledge at the SOTA-level, compared to the recent works [R1, R2]. This demonstrates that our method remains effective also in neuromorphic datasets. The updated results are presented in Table 4 of the manuscript.
> For future work, we plan to propose a method for parameter setting that accounts for gradient sparsity and size.
>
> **Table R3. Accuracy comparison on CIFAR10-DVS with respect to $h$.**
> |Dataset|Architecture|Methods|Time steps|$h$| Acc. (%)|
> |---|---|---|---|--|--|
> |CIFAR10-DVS|VGGSNN|HSD[R1]|5|-|81.10|
> |||TMC[R2]|4|-|81.76|
> |||$A^2SG$|4|0.60|81.68$\pm$0.37|
> ||||4|0.70|81.72$\pm$0.02|
> ||||4|0.75|82.36$\pm$0.01|
> ||||4|0.80|80.29$\pm$0.03|
>
> >W4. TGC is only meaningful when T>1. If configurations such as E-SpikeFormer with $A^2SG$ use T=1, it is difficult to establish the effectiveness of the proposed method.
>
> As the reviewer noted, TGC is generally applicable when the time step is greater than one. However, because spatio-temporal dynamics are an essential property of SNNs, the effective time step in SNNs is typically greater than one. Thus, our approach can be applied to most SNN models.
>
> For I-LIF, as used in E-SpikeFormer, the effective time step is $T$$\times$$D$ rather than $T$. E-SpikeFormer adopted integer LIF neurons with multiple thresholds, which is distinct from the conventional SNNs with LIF neurons. To efficiently train large-scale models on GPUs, it uses integer values as a substitute for the temporal spike trains. Thus, this mechanism implicitly incorporates temporal dynamics even if $T=1$. The model can be regarded as operating with an implicit time step equal to $T$$\times$$D$, rather than a single time step.
>
> Based on this perspective, we aligned the time step of the conventional LIF neurons with the integer activation value of I-LIF. For example, when $T$$\times$$D$ is $1$$\times$$4$ for I-LIF, we applied S-ASG to neurons with an activation value of four, corresponding to the last time step. We then sequentially applied T-ASG to neurons with activation values of three, two, and one.
>
> For the effective window ($\beta$), we set it to be centered on each threshold, as in LIF with a single threshold. For example, if $\textrm{th}_i$ is a threshold at integer $i$, the effective window of $\textrm{th}_i$ is set to $[\textrm{th}_i-\beta_i, \textrm{th}_i+\beta_i]$. From this state, we changed $\beta_i$ through our adaptive method. This is how we implemented our approach in I-LIF neurons. We have added this explanation to the revised manuscript at line 421 (Section 5.3).
>
> Additionally, E-SpikeFormer relies on STE to pass gradients, which can be regarded as a form of surrogate gradient. However, STE is designed primarily for quantization-aware training and does not consider SNN-specific dynamics. In contrast, our approach provides an SNN-aware surrogate gradient. In this regard, our method is also beneficial for I-LIF neurons, which have been widely adopted in SNN-ViT architectures.

---

> > ### Author Response · Authors · 2025-12-02
> >
> > >Q1. Why SGV be measured at the final time step rather than at other time steps?
> >
> > SGV can be measured at any time step. However, we measure SGV at the final time step because the activations are most stable among all time steps. We also provide the performance of $A^2SG$ when SGV is measured at other time steps (two, three, and four). The results of the experiment are shown in Table R4. The experiments were repeated four times, consistent with the experimental conditions in the manuscript. The results show that measuring SGV at time step four yields the best accuracy.
> >
> > **Table R4. Comparison of $A^2SG$ performance with respect to SGV-t.
> > SGV-t denotes the time step at which the SGV is measured.**
> > | Dataset | Architecture | SGV-t | Acc. (%)|
> > | --- | --- | --- | --- |
> > | CIFAR10 | VGG16 | 2 | 94.87$\pm$0.10 |
> > |  |  | 3 | 95.05$\pm$0.11 |
> > |  |  | 4 | 95.29$\pm$0.04 |
> >
> > >Q2. How large is the practical gain from Bayesian search? How much improvement does it deliver over using a fixed $\beta$, a simple grid search, or other methods?
> >
> > The experiments were conducted on VGG16 with CIFAR10, and we report the mean and standard deviation over four runs. The results of the experiment are shown in Table R5. When $\beta$ is fixed to 0.2, training fails to converge. Fixing $\beta$ to 0.5 yields the accuracy that is about 0.25% lower than our method. In the case of grid search, the performance is approximately 0.16% lower than ours. Random search, where $\beta$ is sampled from a uniform distribution over [0,1], performs worse than the fixed ($\beta$=0.5) baseline and shows a performance gap of approximately 0.34% compared to our approach. These results confirm that training performance is significantly affected by the effective window and indicate that an appropriate search strategy for $\beta$ is crucial for training deep SNNs.
> >
> > **Table R5. Comparison of performance for search methods.**
> > | Dataset | Architecture | Methods | Acc. (%)|
> > | --- | --- | --- | --- |
> > | CIFAR10 | VGG16 | fix ($\beta$=0.2) | 21.98$\pm$0.24 |
> > |  |  | fix ($\beta$=0.5) | 95.04$\pm$0.05 |
> > |  |  | grid search | 95.13$\pm$0.05 |
> > |  |  | Random search | 94.95$\pm$0.11 |
> > |  |  | $A^2SG$ | 95.29$\pm$0.05 |
> >
> > We thank the reviewer once again for the valuable comments, and we hope that our responses adequately address the concerns and misunderstandings.
> >
> > [R1] Zhong, Xian, et al. "Towards low-latency event-based visual recognition with hybrid step-wise distillation spiking neural networks." *Proceedings of the 32nd ACM international conference on multimedia*. 2024.
> >
> > [R2] Jiaqi Yan, Changping Wang, De Ma, Huajin Tang, Qian Zheng, and Gang Pan. Training high performance spiking neural network by temporal model  calibration. In Forty-second International Conference on Machine Learning, 2025.

---

### Meta-Review · Area_Chair_fzbU · 2025-12-23

**Summary:**

The paper addresses the challenges of training deep Spiking Neural Networks (SNNs), specifically focusing on sharp loss landscapes and temporal inconsistency caused by surrogate gradients. The authors propose a unified framework called $A^2SG$, which utilizes adaptive gradients to adjust the effective window for spatio-temporal adaptation and asymmetric gradients to reflect neuronal dynamics.

The reviewers generally acknowledged the theoretical novelty and the logical soundness of linking gradient variation to loss landscape curvature. However, the initial consensus was borderline (mostly scores of 4), primarily due to concerns regarding the unreported computational overhead of the Bayesian search, questions about the validity of theoretical assumptions (Gaussian distributions), and skepticism regarding the magnitude of performance gains on neuromorphic datasets (CIFAR10-DVS) and large-scale task.

Overall, I think the authors provided a comprehensive rebuttal, including new experimental data on computational overhead, sensitivity analyses, and improved results on neuromorphic datasets. Yet the consensus feedback from the reviews tends to be on the borderline. Given the competitive pool for this year's ICLR, I would suggest a reject.

**Reviewer Concerns:**

### **Addressed Concerns**
**Computational Overhead:** Multiple reviewers (WK4e, uKH8, wBJ2) flagged the lack of analysis regarding the cost of the Bayesian optimization search run every epoch. The authors addressed this by providing a detailed overhead analysis (Table R2), showing a manageable increase in training time of approximately 11-15%.

**Performance on Neuromorphic Data:** Reviewers WK4e and YJia noted that the gains on CIFAR10-DVS were marginal (<1%). The authors admitted an oversight in hyperparameter settings for this dataset, performed a grid search during the rebuttal, and reported a new accuracy of 82.36%, representing a >1% improvement over the baseline and achieving state-of-the-art levels.

**Theoretical Assumptions:** Reviewers WK4e and wBJ2 questioned the assumption of Gaussian membrane potentials and the validity of the linear approximation. The authors provided empirical data (Table R1/R2) tracking $L\kappa$ and $\sigma^2$ across epochs, demonstrating that the necessary conditions for their theorem are satisfied as training progresses.

**Sensitivity Analysis:** Reviewers uKH8 and wBJ2 requested sensitivity analyses for the new hyperparameters ($n_{obs}$, update frequency). The authors included a new section (5.7) and Table R3/R4 showing that the method is robust to changes in update frequency and search parameters.

### **Remaining Concerns**

**Large-Scale Validation (ImageNet):** Reviewer YJia noted that improvements on ImageNet were small (<1%) and questioned the baseline model choice. While the authors defended their choice and methodology, they acknowledged they could not complete additional large-scale experiments on simpler architectures (like ResNet on ImageNet) during the discussion period due to resource constraints.

**Incremental Contribution:** Reviewer YJia maintained that surrogate gradient improvement is only one optimization route and felt the overall contribution was limited, despite the technical rebuttals.

**Reviewer Scores:**

**Reviewer WK4e (Current: 4 $\rightarrow$ Predicted: 5/6):**

**Rationale:** This reviewer stated they would raise their score if weaknesses were addressed. The authors successfully provided the missing overhead data, explained the theoretical assumptions with empirical backing, and improved the neuromorphic results significantly.

**Reviewer uKH8 (Current: 4 $\rightarrow$ Predicted: 5):**

**Rationale:** The authors addressed this reviewer's main concerns regarding overhead, sensitivity, and applicability to other neuron models (PLIF). While the reviewer rated the presentation "fair," the technical clarifications should warrant a score increase.

**Reviewer YJia (Current: 4 $\rightarrow$ Predicted: 4):**

**Rationale: **This reviewer explicitly commented after reading the rebuttal: "I tend to maintain my current rating". Despite the authors improving the DVS results to meet the reviewer's implicit threshold (>1%), the reviewer appears unconvinced by the overall significance of the contribution.

**Reviewer wBJ2 (Current: 6 $\rightarrow$ Predicted: 7): **
**Rationale: ** This reviewer was already positive. The authors addressed their "major flaw" (overhead omission) and provided the requested CIFAR-100 ablation. This solidifies the paper's standing significantly for this reviewer.

---

### Decision · Program_Chairs · 2026-01-26

Reject